# Efficient Activation Function Optimization through Surrogate Modeling

**Garrett Bingham**[*]
The University of Texas at Austin
and Cognizant AI Labs
San Francisco, CA 94105
garrett@gjb.ai

**Risto Miikkulainen**
The University of Texas at Austin
and Cognizant AI Labs
San Francisco, CA 94105
risto@cs.utexas.edu

## Abstract

Carefully designed activation functions can improve the performance of neural networks in many machine learning tasks. However, it is difficult for humans to construct optimal activation functions, and current activation function search algorithms are prohibitively expensive. This paper aims to improve the state of the art through three steps: First, the benchmark datasets `Act-Bench-CNN`, `Act-Bench-ResNet`, and `Act-Bench-ViT` were created by training convolutional, residual, and vision transformer architectures from scratch with 2,913 systematically generated activation functions. Second, a characterization of the benchmark space was developed, leading to a new surrogate-based method for optimization. More specifically, the spectrum of the Fisher information matrix associated with the model's predictive distribution at initialization and the activation function's output distribution were found to be highly predictive of performance. Third, the surrogate was used to discover improved activation functions in several real-world tasks, with a surprising finding: a sigmoidal design that outperformed all other activation functions was discovered, challenging the status quo of always using rectifier nonlinearities in deep learning. Each of these steps is a contribution in its own right; together they serve as a practical and theoretical foundation for further research on activation function optimization.

## 1 Introduction

Activation functions are an important choice in neural network design [2, 46]. In order to realize the benefits of good activation functions, researchers often design new functions based on characteristics like smoothness, groundedness, monotonicity, and limit behavior. While these properties have proven useful, humans are ultimately limited by design biases and by the relatively small number of functions they can consider. On the other hand, automated search methods can evaluate thousands of unique functions, and as a result, often discover better activation functions than those designed by humans. However, such approaches do not usually have a theoretical justification, and instead focus only on performance. This limitation results in computationally inefficient ad hoc algorithms that may miss good solutions and may not scale to large models and datasets.

This paper addresses these drawbacks in a data-driven way through three steps. First, in order to provide a foundation for theory and algorithm development, convolutional, residual, and vision transformer based architectures were trained from scratch with 2,913 different activation functions, resulting in three activation function benchmark datasets: `Act-Bench-CNN`, `Act-Bench-ResNet`,

---

[*]GB is currently a research scientist at Google DeepMind. AQuaSurF code is available at https://github.com/cognizant-ai-labs/aquasurf, and the benchmark datasets are at https://github.com/cognizant-ai-labs/act-bench.

37th Conference on Neural Information Processing Systems (NeurIPS 2023).

and `Act-Bench-ViT`. These datasets make it possible to analyze activation function properties at a large scale in order to determine which are most predictive of performance.

The second step was to characterize the activation functions in these benchmark datasets analytically, leading to a surrogate performance measure. Exploratory data analysis revealed two activation function properties that are highly indicative of performance: (1) the spectrum of the Fisher information matrix associated with the model's predictive distribution at initialization, and (2) the activation function's output distribution. Both sets of features contribute unique information. Both are predictive of performance on their own, but they are most powerful when used in tandem. These features were combined to create a metric space where a low-dimensional representation of the activation functions was learned. This space was then used as a surrogate in the search for good activation functions.

In the third step, this surrogate was evaluated experimentally, first by verifying that it can discover known good functions in the benchmark datasets efficiently and reliably, and second by demonstrating that it can discover improved activation functions in new tasks involving different datasets, search spaces, and architectures. The representation turned out to be so powerful that an out-of-the-box regression algorithm was able to search it effectively. This algorithm improved performance on various tasks, and also discovered a sigmoidal activation function that outperformed all baselines, a surprising discovery that challenges the common practice of using ReLU and its variants. The approach, called AQuaSurF (Activation Quality with a Surrogate Function), is orders of magnitude more efficient than past work. Indeed, whereas previous approaches evaluated hundreds or thousands of activation functions, AQuaSurF requires only tens of evaluations in order to discover functions that outperform a wide range of baseline activation functions in each context. Code implementing the AQuaSurF algorithm is available at `https://github.com/cognizant-ai-labs/aquasurf`.

Prior research on activation function optimization and Fisher information matrices is reviewed in Section A. This work extends it in three ways. First, the benchmark collections are made available at `https://github.com/cognizant-ai-labs/act-bench`, providing a foundation for further research on activation function optimization. Second, the low-dimensional representation of the Fisher information matrix makes it a practical surrogate measure, making it possible to apply it to not only activation function design, but potentially also to other applications in the future. Third, the already-discovered functions can be used immediately to improve performance in image processing tasks, and potentially in other tasks in the future.

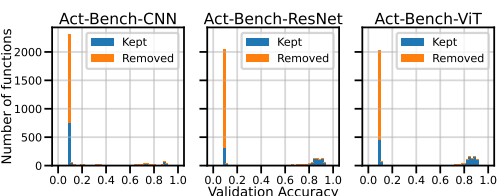

Figure 1: Distribution of validation accuracies with 2,913 unique activation functions from the three benchmark datasets. Many activation functions result in failed training (indicated by the chance accuracy of 0.1), suggesting that searching for activation functions is a challenging problem. However, most of these functions have invalid FIM eigenvalues, and can thus be filtered out effectively.

## 2 Activation Function Benchmarks

As the first step, three activation function benchmark datasets are introduced: `Act-Bench-CNN`, `Act-Bench-ResNet`, and `Act-Bench-ViT`. Each dataset contains training results for 2,913 unique activation functions when paired with different architectures and tasks: All-CNN-C on CIFAR-10, ResNet-56 on CIFAR-10, and MobileViTv2-0.5 on Imagenette [22, 24, 31, 41, 51]. These functions were created using the main three-node computation graph from PANGAEA [5]. Details are in Appendix B.

Figure 1 shows the distribution of validation accuracies in these datasets. In all three datasets, the distribution is highly skewed towards functions that result in failed training. The plots suggest that it is difficult to design good activation functions, and explain why existing methods are computationally expensive. Notwithstanding this difficulty, the histograms show that many unique functions do achieve good performance. Thus, searching for new activation functions is a worthwhile task that requires a smart approach.

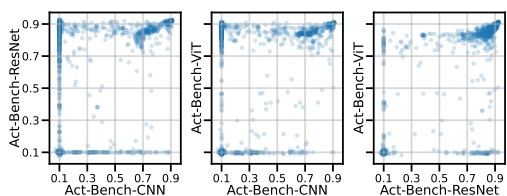

Figure 2: Distribution of validation accuracies across the benchmark datasets. Each point represents a unique activation function's performance on two of the three datasets. Some functions perform well on all tasks, while others are specialized.

Figure 2 shows the same data as Figure 1, but with scatter plots that show how performance varies across different tasks. All three plots contain linearly correlated clusters of points in the upper right corner, suggesting that there are modifications to activation functions that make them more powerful across tasks. However, the best results come from discovering functions specialized to individual tasks, indicated by the clusters of points in the upper left and lower right corners.

The three benchmark datasets form a foundation for developing and evaluating methods for automated activation function design. In the next two sections, they are used to develop a surrogate performance metric, making it possible to scale up activation function optimization to large networks and datasets.

## 3 Features and Distance Metrics

To make efficient search for activation functions possible, the surrogate space needs to be low-dimensional, represent informative features, and have an appropriate distance metric. In the second step, an approach is developed based on (1) the eigenvalues of the Fisher information matrix and (2) the outputs of the activation function. This section motivates each feature type and develops a metric for computing distances between activation functions. They form a surrogate in the next section.

**FIM Eigenvalues**   The Fisher information matrix (FIM) is an important concept in characterizing neural network models. Viewed from various perspectives, the FIM determines a neural network's capacity for learning, ability to generalize, the robustness of the network to small perturbations of its parameters, and the geometry of the loss function near the global minimum [16, 21, 27–29, 33, 34].

Consider a neural network $f$ with weights $\boldsymbol{\theta}$. Given inputs $\mathbf{x}$ drawn from a training distribution $Q_{\mathbf{x}}$, the network defines the conditional distribution $R_{\mathbf{y}|f(\mathbf{x};\boldsymbol{\theta})}$. The FIM associated with this model is

$$\mathbf{F} = \mathbb{E}_{\substack{\mathbf{x} \sim Q_{\mathbf{x}} \\ \mathbf{y} \sim R_{\mathbf{y}|f(\mathbf{x};\boldsymbol{\theta})}}} \left[ \nabla_{\boldsymbol{\theta}} \mathcal{L}(\mathbf{y}, f(\mathbf{x};\boldsymbol{\theta})) \nabla_{\boldsymbol{\theta}} \mathcal{L}(\mathbf{y}, f(\mathbf{x};\boldsymbol{\theta}))^{\top} \right], \tag{1}$$

where $\mathcal{L}(\mathbf{y}, \mathbf{z})$ is the loss function representing the negative log-likelihood associated with $R_{\mathbf{y}|f(\mathbf{x};\boldsymbol{\theta})}$.

The FIM has $|\boldsymbol{\theta}|$ eigenvalues. The distribution of eigenvalues can be represented by binning the eigenvalues into an $m$-bucket histogram, and this $m$-dimensional vector serves as a computational characterization of the network. To calculate the FIM and its eigenvalues, this paper uses the K-FAC approach [20, 38]. Full details are in Appendix C.

Different activation functions induce different FIM eigenvalues for a given neural network. They can be calculated at initialization without training; they can thus serve as a low-dimensional feature vector representation of the activation function. The FIM eigenvalues are immediately useful for filtering out poor activation functions; if they are invalid, the activation function is likely to fail in training (Figure 1). However, in order to use them as a surrogate, a distance metric needs to be defined.

Given a neural network architecture $f$, let $f_{\phi}$ and $f_{\psi}$ be two instantiations with different activation functions $\phi$ and $\psi$. Let $\mu_l$ and $\nu_l$ represent the distributions of eigenvalues corresponding to the weights in layer $l$ of neural networks $f_{\phi}$ and $f_{\psi}$, respectively, and let $w_l$ be the number of weights in layer $l$ of the networks. The distance between $f_{\phi}$ and $f_{\psi}$ is then computed as a weighted layer-wise sum of 1-Wasserstein distances

$$d(f_{\phi}, f_{\psi}) = \sum_{l=1}^{L} W_1(\mu_l, \nu_l)/w_l. \tag{2}$$

With this distance metric, the FIM eigenvalue vector representations encode a low-dimensional embedding space for activation functions, making efficient search possible. Because the FIM eigenvalues depend on several factors (Equation 1), including the activation function $\phi$, network architecture $f$, data distribution $Q$, and loss function $\mathcal{L}$, they are susceptible to more potential sources of noise. Fortunately, incorporating activation function outputs helps to compensate for this noise.

**Activation Function Outputs**   The shape of an activation function $\psi$ can be described by a vector of $n$ sample values $\psi(x)$. If the network's weights are appropriately initialized, the input activations to its neurons are initially distributed as $\mathcal{N}(0, 1)$ [4]. Therefore, the sampling $x \sim \mathcal{N}(0, 1)$ provides an $n$-dimensional feature vector that represents the expected use of the activation function at initialization. A distance metric in this feature vector space can be defined naturally as the Euclidean distance

$$d(f_{\phi}, f_{\psi}) = \sqrt{\sum_{i=1}^{n} (\phi(x_i) - \psi(x_i))^2/n}, \quad x \sim \mathcal{N}(0, 1). \tag{3}$$

Functions with similar shapes will have a small distance between them, while those with different shapes will have a large distance. Because these output feature vectors depend only on the activation function, they are reliable and inexpensive to compute. Most importantly, together with the FIM eigenvalues, they constitute a powerful surrogate search space, demonstrated in the next section.

# 4 Using the Features as a Surrogate

In this section, the UMAP dimensionality reduction technique is used to visualize the FIM and output features across the benchmark datasets. This visualization leads to a combined surrogate space that can be used to accelerate the search for good activation functions.

**Visualization with UMAP** The features developed above can be visualized using the UMAP algorithm [40]. UMAP is a dimension reduction approach similar to t-SNE, but is better at scaling to large sample sizes and preserving global structure [55]. As a first demonstration, Figure 3 shows a 2D representation of the 2,913 activation functions in the benchmark datasets. Each function was represented as an 80-dimensional vector of output values. Interpolating between embedded points confirms that UMAP learns a good underlying representation.

UMAP was also used to project the activation functions to nine two-dimensional spaces according to the distance metrics in Equations 2 and 3. In Figure 4, each column represents a different benchmark dataset (`Act-Bench-CNN`, `Act-Bench-ResNet`, or `Act-Bench-ViT`) and each row a different distance metric (FIM eigenvalues with $m = \lfloor |\boldsymbol{\theta}|/100 \rfloor$, activation function outputs with $n = 1,000$, or both). The plots only show activation functions that were not filtered out. Each point represents a unique function, colored according to its validation accuracy on the benchmark task. Although the performance of each activation function is already known, this information was not given to UMAP; the embeddings are entirely unsupervised.

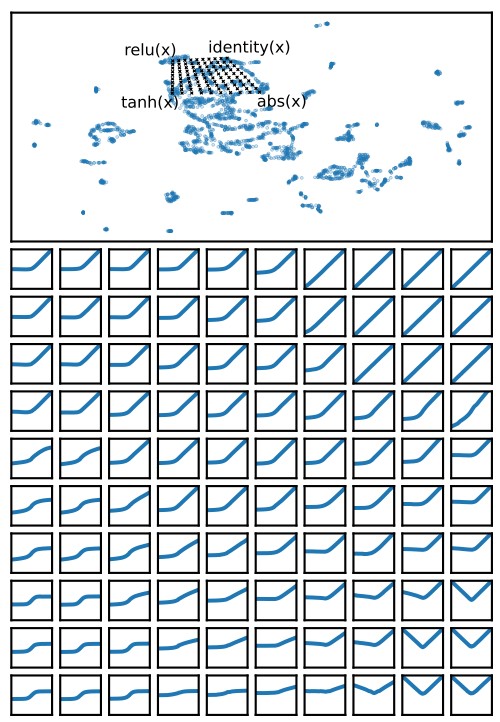

Figure 3: UMAP embedding of the 2,913 activation functions in the benchmark datasets. Each point stands for a unique activation function, represented by an 80-dimensional output feature vector. The embedding locations of four common activation functions are labeled. The black x's mark coordinates interpolating between these four functions, and the grid of plots on the bottom shows reconstructed activation functions at each of these points. UMAP interpolates smoothly between different kinds of functions, suggesting that it is a good approach for learning low-dimensional representations of activation functions.

Thus, Figure 4 illustrates how predictive each feature type is of activation function performance in each dataset. The next subsections evaluate each feature type in this role in detail, and show that utilizing both features provides better results than either feature alone. Details are in Appendix D.

**FIM Eigenvalues** The first row of Figure 4 shows the 2D UMAP embeddings of the FIM eigenvalue vectors associated with each activation function. There are clusters in these plots where the points share similar colors, indicating distinct activation functions with similar FIM eigenvalues. Such functions induce similar training dynamics in the neural network and lead to similar performance. On the other hand, some clusters contain activation functions with a wide range of performances, and some points do not belong to any cluster at all. Overall, the plots suggest that FIM eigenvalues are a useful predictor of performance, but that incorporating additional features could lead to better results.

**Activation Function Outputs** The middle row of Figure 4 shows the 2D UMAP embeddings of the output vectors associated with each activation function. Points are close to each other in this space if the corresponding activation functions have similar shapes. These plots are demonstrably

more informative than the plots based on the FIM eigenvalues in three ways. First, the purple points are better separated from the others. This separation means that activation functions that fail (those achieving 0.1 chance accuracy) are better separated from those that do well. Second, most points' immediate neighbors have similar colors. This similarity means that activation functions with similar shapes lead to similar accuracy, and analyzing activation function outputs on their own is more informative than analyzing the FIM eigenvalues. Third, the plots include multiple regions where there are one-dimensional manifolds that exhibit smooth transitions in accuracy, from purple to blue to green to yellow. Thus, not only does UMAP successfully embed similar activation functions near each other, but it also is able to organize the activation functions in a meaningful way.

There is one drawback to this approach: the performant activation functions (those represented by yellow dots) are often in distinct clusters. This dispersion means that a search algorithm would have to explore multiple areas of the search space in order to find all of the best functions. As the next subsection suggests, this issue can be alleviated by utilizing both FIM eigenvalues and activation function outputs.

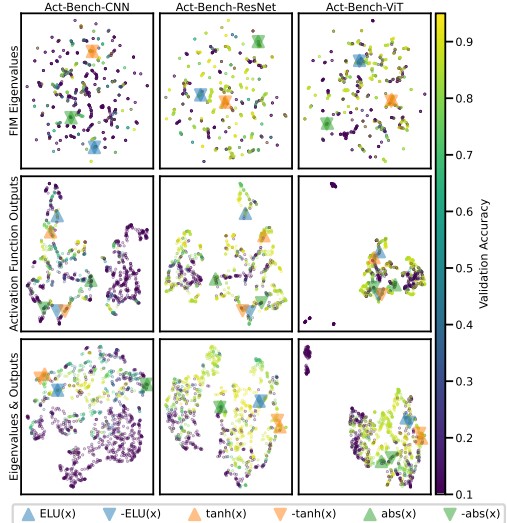

Figure 4: UMAP embeddings of activation functions for each dataset (column) and feature type (row). Each point represents a unique activation function; the points are colored by validation accuracy on the given dataset. The colored triangles identify the locations of six well-known activation functions. The areas of similar performance are more continuous in the bottom row; that is, using both FIM eigenvalues and activation function outputs provides a better low-dimensional representation than either feature alone.

**Combining Eigenvalues & Outputs** The UMAP algorithm uses an intermediate fuzzy topological representation to represent relationships between data points, similar to a neighborhood graph. This property makes it possible to combine multiple sources of data by taking intersections or unions of the representations in order to yield new representations [40]. The bottom row of Figure 4 utilizes both FIM eigenvalues and activation function outputs by taking the union of the two representations. Thus, activation functions are embedded close to each other in this space if they have similar shapes, if they induce similar FIM eigenvalues, or both.

The bottom row of Figure 4 shows the benefits of combining the two features. Unlike the activation function output plots, which contain multiple clusters of high-performing activation functions in different locations in the embedding space, the combined UMAP model embeds all of the best activation functions in similar regions. The combined UMAP model also places poor activation functions (purple points) in the edge of the embedding space, and brings good functions (yellow points) to the center. Thus, the embedding space is more convex, and therefore easier to optimize.

In general, activation functions with similar shapes lead to similar performances, and those with different shapes often produce different results. This property is why the middle row of Figure 4 appears locally smooth. However, in some cases the shape of the activation function does not tell the whole story, and additional information is needed to ascertain its performance.

For example, the colored triangles in Figure 4 identify the location of six activation functions in the low-dimensional space. In the activation function output space (middle row), all of these functions are mapped to different regions of the space. The points are spread apart because an activation function and its negative have very different shapes, i.e. their output will be different for every nonzero input (Figure 5). In contrast, in the FIM eigenvalue space (top row), the points for these pairs of functions overlap because the FIM eigenvalues are comparable (Figure 5). Indeed, assuming the weights are initialized from a distribution symmetric about zero, negating an activation function does not change the training dynamics of a neural network, and they are functionally equivalent.

This issue complicates the search process in two ways. First, good activation functions are mapped to different regions of the embedding space, and so a search algorithm must explore multiple areas in order to find the best function. Second, distinct regions of the space may contain redundant

information: if ELU($x$) is known to be a good activation function, it is not helpful to spend compute resources evaluating $-$ELU($x$) only to discover that it achieves the same performance.

Negating an activation function is a clear example of a modification that changes the shape of the activation function, but does not affect the training of a neural network. More broadly, it is likely that there exist activation functions that differ in other ways (besides just negation), but that still induce similar training dynamics in neural networks. Fortunately, utilizing FIM eigenvalues and activation function outputs together provides enough information to tease out these relationships. FIM eigenvalues take into account the activation function, the neural network architecture, the loss function, and the data distribution. The eigenvalues are more meaningful features than activation function outputs, which only depend on the shape of the function. However, as Figure 4 shows, the FIM eigenvalues are noisier features, while the activation function outputs are quite reliable. Thus, utilizing both features is a natural way to combine their strengths and address their weaknesses.

**Constructing a Surrogate**  These observations suggest an opportunity for an effective surrogate measure: The UMAP coordinates in the bottom row of Figure 4 have the information needed to predict how well an activation function will perform. They capture the essence of the $m$ and $n$ dimensional feature vectors, and distill it into a 2D representation that can be computed efficiently and used to guide the search for good functions. As the third step in this research, the next two sections evaluate this process experimentally, demonstrating that it is efficient and reliable, and that it scales to new and challenging datasets and search spaces.

## 5   Searching on the Benchmarks

Searching for activation functions typically requires training a neural network from scratch in order to evaluate each candidate function fully, which is often computationally expensive. With the benchmark datasets, the results are already precomputed. This information makes it possible to experiment with different search algorithms and conduct repeated trials to understand the statistical significance of the results. These results serve to inform both algorithm design and feature selection, as shown in this section.

**Setup**  Three algorithms were evaluated: weighted $k$-nearest regression with $k = 3$ (KNR), random forest regression (RFR), and

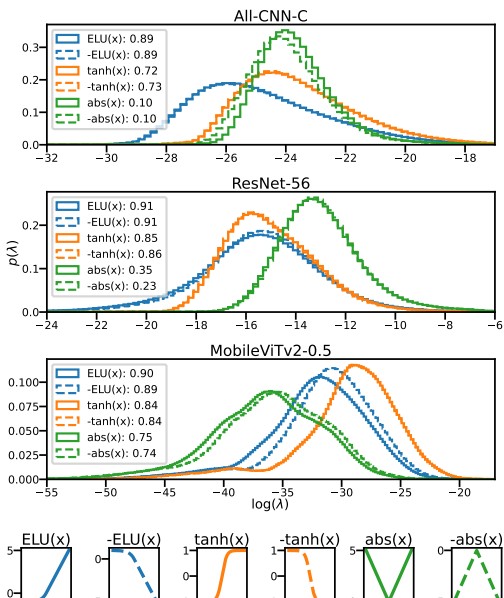

Figure 5: FIM eigenvalue distributions for different architectures and activation functions. The legends show the activation function and the corresponding validation accuracy in different tasks. Although negating an activation function changes its shape, it does not substantially change its behavior nor its performance. FIM eigenvalues capture this relationship between activation functions. The eigenvalues are thus useful for finding activation functions that appear different but in fact behave similarly, and these discoveries in turn improve the efficiency of activation function search.

support vector regression (SVR). Gaussian Process Regression (GPR) was also evaluated but found to be inconsistent in preliminary experiments (Appendix E). Random search (RS) was included as a baseline comparison; it did not utilize the FIM eigenvalue filtering mechanism. The algorithms were used out of the box with default hyperparameters from the scikit-learn package [47]. They were provided different activation function features in order to understand their potential to predict performance. The features included FIM eigenvalues, activation function outputs, or both. The features were preprocessed and embedded in a two-dimensional space by UMAP. These representations are visualized in Figure 4; the coordinates of each point correspond exactly to the information given to the regression algorithms.

The ReLU activation function is ubiquitous in machine learning. For many neural network architectures, the performance with ReLU is already known [2, 45, 46], which makes it a good starting point for search. For this reason, the search algorithms began by evaluating ReLU and seven other

randomly chosen activation functions. In general, such evaluation requires training from scratch, but with the benchmark datasets, it requires only looking up the precomputed results. The algorithms then used the validation accuracy of these eight functions to predict the performance of all unevaluated functions in the dataset. The activation function with the highest predicted accuracy was then evaluated. The performance of this new function was then added to the list of known results, and this process continued until 100 activation functions had been evaluated. Each experiment comprising a different search algorithm, activation function feature set, and benchmark dataset was repeated 100 times. Full details are in Appendix E.

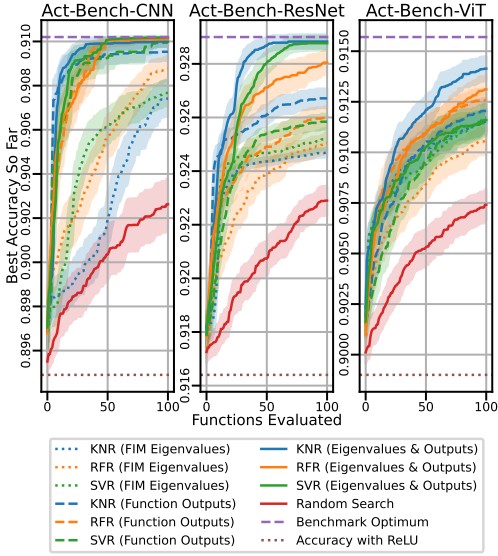

**Results**  Figure 6 shows the results of the searches. Importantly, the curves do not depict just one search trial. Instead, they represent the average performance aggregated from 100 independent runs, which is made possible by the benchmark datasets. As indicated by the shaded confidence intervals, the results are reliable and are not simply due to chance.

A number of conclusions can be drawn from Figure 6. First, all search algorithms, even random search, reliably discover activation functions that outperform ReLU. This finding is supported by previous work (reviewed in Section A): Although ReLU is a good activation function that performs well in many different tasks, better performance can be achieved with novel activation functions. Therefore, continuing to use ReLU in the future is unlikely to lead to best results; The choice of the activation function should be an important part of the design, similar to the choice of the network architecture or the selection of its hyperparameters.

Second, all regression algorithms outperform random search. This finding holds across the three types of activation function features and across the three benchmark datasets. The FIM eigenvalues and activation function outputs are thus important in predicting performance of activation functions.

Third, regression algorithms trained on both FIM eigenvalues and activation function outputs outperform algorithms trained on just eigenvalues or outputs alone. This result is consistent across the regression algorithms and benchmark datasets. It suggests that the FIM eigenvalues and activation function outputs contribute complimentary pieces of information. The finding quantitatively reinforces the qualitative visual-

Figure 6: Search results on the three benchmark datasets. Each curve represents a different search algorithm (KNR, RFR, or SVR) utilizing a different UMAP feature (FIM eigenvalues, function outputs, or both; these features are visualized in Figure 4). The curves represent the validation accuracy of the best activation function discovered so far, averaged across 100 independent trials, and the shaded areas show the 95% confidence interval around the mean. In all cases, regression with UMAP features outperforms random search, and searching with both eigenvalues and outputs outperforms searching with either feature alone. Of the three regression algorithms, KNR performs the best, rapidly surpassing ReLU and quickly discovering near-optimal activation functions in all benchmark tasks. Thus, the features make it possible to find good activation functions efficiently and reliably even with off-the-shelf search methods; the benchmark datasets make it possible to demonstrate these conclusions with statistical reliability.

ization in Figure 4: FIM eigenvalues are useful for matching activation functions that induce similar training dynamics in neural networks, activation function outputs enable a low-dimensional representation where search is more practical, and combining the two features results in a problem that is more convex and easier to optimize.

Fourth, the searches are efficient. Previous approaches require hundreds or thousands of evaluations to discover good activation functions [5, 6, 49]. In contrast, this paper leverages FIM eigenvalues and activation function outputs to reduce the problem to simple two-dimensional regression; the features are powerful enough that out-of-the-box regression algorithms can discover good functions with only tens of evaluations. This efficiency makes it possible to search for better functions directly on large datasets such as ImageNet [11], demonstrated next.

# 6 Searching with New Settings

The experiments in Section 5 used precomputed datasets and search spaces to demonstrate that UMAP embeddings are predictive of activation function performance, and that KNR can find good functions based on them. To verify that these conclusions extend beyond the benchmark tasks, this section contains three experiments demonstrating that AQuaSurF scales up to more challenging datasets and search spaces, that the activation functions can be transferred to other tasks, and that AQuaSurF extends to new architectures and baseline functions.

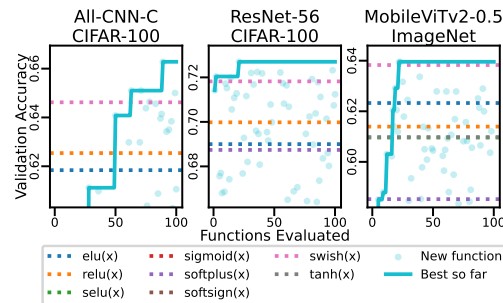

Figure 7: Progress of activation function searches. Each point represents the validation accuracy with a unique activation function, and the solid line indicates the performance of the best activation function found so far. AQuaSurF discovers new activation functions that outperform all baseline functions in every case.

**Scaling Up the Datasets and Search Space** In the first experiment, the tasks involve larger and more challenging datasets: All-CNN-C on CIFAR-100, ResNet-56 on CIFAR-100, and MobileViTv2-0.5 on ImageNet. Additionally, a larger space with 425,896 unique activation functions was searched, based on four-node computation graphs (Appendix B). This space is large, diverse, and not precomputed, putting the conclusions from the benchmark experiments to test in a production setting.

Based on the benchmark results, KNR with $k = 3$ was used as the search algorithm. The searches all began by evaluating the same eight existing activation functions: ELU, ReLU, SELU, sigmoid, Softplus, Softsign, Swish, and tanh. From this starting point, eight workers operated in parallel evaluating the functions with the highest predicted performance. Details are in Appendix E.

Figure 7 shows that all three searches find improved activation functions over time, and Figure 10 in Appendix B shows how the searches navigate the search space. In every experiment, new activation functions were discovered that outperform all baseline functions. Although the search space is large, the searches are efficient, requiring only tens of evaluations to improve performance. Impressively, the search with ResNet-56 on CIFAR-100 produced an activation function that outperformed all baselines on just the second evaluation.

Table 1 shows the final results from AQuaSurF. The results reinforce the fact that substantial gains can be obtained when using better activation functions than the default ReLU, and especially those optimized specifically for the task.

**Transferring to a New Task** In the second experiment, the best activation functions from Table 1 were transferred to a new task: ResNet-50 on ImageNet. As demonstrated in Table 2, good functions can be discovered efficiently in smaller tasks and then used to improve performance in larger ones.

**New Architectures and Baseline Functions** In the third experiment, the CoAtNet architecture was trained on Imagenette [10]. As a hybrid convolution and attention architecture, CoAtNet presents a new challenge for AQuaSurF. The activation functions ELiSH, GELU, HardSigmoid, Leaky ReLU, and Mish [3, 23, 37, 43] were added to the original set of baseline functions (Table 1), as well as to the set of `unary` operators, forming a new search space for AQuaSurF to explore (Appendix B).

The results show that AQuaSurF extends to architectures and baseline functions not considered in the benchmark tasks (Table 3). AQuaSurF discovered multiple activation functions that substantially outperform all baseline functions. Although the extended list of baseline functions presents a more challenging task, it also provides the surrogate function more information that it uses for performance prediction, resulting in the discovery of even better functions.

# 7 Understanding the Discoveries

Aside from the raw performance improvements afforded by AQuaSurF, the experiments on the new settings are particularly interesting because they illustrate both the process of refining existing activation functions and the process of discovering novel designs.

Table 1: Accuracy with different activation functions. The CIFAR-100 results show the median test accuracy from three runs, and the ImageNet results show the validation accuracy from a single run. AQuaSurF discovers novel activation functions that outperform all baselines in every case. This result demonstrates both that good functions matter, and the power of optimizing them to the task.

| All-CNN-C on CIFAR-100 | | ResNet-56 on CIFAR-100 | | MobileViTv2-0.5 on ImageNet | |
|---|---|---|---|---|---|
| HardSigmoid(HardSigmoid$(x)$) $\cdot$ ELU$(x)$ | **0.6990** | Swish$(-2x)$ | **0.7469** | $-x \cdot \sigma(x) \cdot$ HardSigmoid$(x)$ | **0.6396** |
| $\sigma$(Softsign$(x)$) $\cdot$ ELU$(x)$ | 0.6950 | SELU$(\sinh(e^{\arctan(x)} - 1))$ | 0.7458 | ELU(Swish$(-x)$) | 0.6394 |
| Swish$(x)$/SELU$(1)$ | 0.6931 | $x \cdot$ erfc(ELU$(x)$) | 0.7419 | Swish$(x) \cdot$ erfc(bessel_i0e$(x)$) | 0.6336 |
| ELU | 0.6312 | ELU | 0.7411 | ELU | 0.6233 |
| ReLU | 0.6897 | ReLU | 0.7348 | ReLU | 0.6139 |
| SELU | 0.0100 | SELU | 0.6967 | SELU | 0.6096 |
| sigmoid | 0.0100 | sigmoid | 0.5766 | sigmoid | 0.5032 |
| Softplus | 0.6563 | Softplus | 0.7397 | Softplus | 0.5853 |
| Softsign | 0.2570 | Softsign | 0.6624 | Softsign | 0.5710 |
| Swish | 0.6913 | Swish | 0.7401 | Swish | 0.6383 |
| tanh | 0.3757 | tanh | 0.6754 | tanh | 0.6098 |

**Refinement and Novelty** Figure 8 shows different activation functions discovered during the searches. (Plots of all 100 functions evaluated in each search are included in Figures 13–16 in Appendix E.) Visually, many the best functions (shown in 8a) are similar to existing functions like ELU and Swish, with subtle changes in their saturation value, the slope of the positive segment, and the width and depth of the negative bump. This result is not surprising since these functions formed the starting point for the search. Indeed, after a few good functions were found, much of the search process focused on refining their design (Figure 10 in Appendix B). Although these refinements appear small, they were not known ahead of time and they are significant, as evidenced by the final results (Tables 1–3).

However, some of the best discovered activation functions, including the top function for the CoAtNet experiment, employ properties uncommon among the usual deep learning activation functions (Figure 8b): Some of them have discontinuous derivatives at $x = 0$; some do not saturate, but diverge as $x \to \pm\infty$; some of them contain positive bumps (in contrast to e.g. Swish, which features a negative bump). Many of these functions performed comparably to the existing best functions, and all of them outperformed ReLU. In the future, these designs may provide a comprehensive foundation for discovering better activation functions for specific new tasks.

Together, the plots show that AQuaSurF is capable of both exploitation (Figure 8a) and exploration (Figure 8b). In the future, it will be interesting to explore tradeoffs between these concepts. A more comprehensive discussion of this and other future research directions is included in Appendix F.

**Discovering a Hybrid Rectifier-Sigmoidal Activation Function** In the past, sigmoidal nonlinearities like sigmoid and

Table 2: ResNet-50 top-1 accuracy on ImageNet. Results are the median of three runs. The best activation functions discovered in the searches (Table 1) successfully transfer to this new task, with eight of the nine functions outperforming ReLU.

| | |
|---|---|
| $-x \cdot \sigma(x) \cdot$ HardSigmoid$(x)$ | **0.7776** |
| Swish$(x)$/SELU$(1)$ | 0.7771 |
| Swish$(x) \cdot$ erfc(bessel_i0e$(x)$) | 0.7755 |
| $\sigma$(Softsign$(x)$) $\cdot$ ELU$(x)$ | 0.7734 |
| SELU$(\sinh(e^{\arctan(x)} - 1))$ | 0.7719 |
| HardSigmoid(HardSigmoid$(x)$) $\cdot$ ELU$(x)$ | 0.7718 |
| ELU(Swish$(-x)$) | 0.7679 |
| Swish$(-2x)$ | 0.7664 |
| $x \cdot$ erfc(ELU$(x)$) | 0.7635 |
| ReLU$(x)$ | 0.7660 |

Table 3: CoAtNet validation accuracy on Imagenette. AQuaSurF finds novel functions that outperform all baselines.

| | |
|---|---|
| erfc(Softplus$(x))^2$ | **0.8907** |
| min$\{$Softplus$(x)^2, -x\}$ | 0.8861 |
| arcsinh(ELU(Swish$(x)$)) | 0.8828 |
| ELiSH | 0.1000 |
| ELU | 0.8629 |
| GELU | 0.8841 |
| HardSigmoid | 0.8487 |
| Leaky ReLU | 0.8815 |
| Mish | 0.8762 |
| ReLU | 0.8772 |
| SELU | 0.8194 |
| sigmoid | 0.8586 |
| Softplus | 0.8678 |
| Softsign | 0.8530 |
| Swish | 0.8736 |
| tanh | 0.8415 |

tanh were often used because they saturate and thus prevent exploding signals. However, currently these functions are usually discarded in favor of rectifier nonlinearities like ReLU and its variants as these functions give better performance on modern deep learning benchmarks [2]. Indeed, in Tables 1 and 3, sigmoid, tanh, HardSigmoid, and Softsign all perform relatively poorly. It is therefore surprising to see that the very best function discovered in the CoAtNet experiment, erfc(Softplus$(x))^2$ (bottom left of Figure 8a), is sigmoidal in shape.

Why does this function perform so well? As shown in Figure 9, the function saturates to 1 as $x \rightarrow -\infty$ and to 0 as $x \rightarrow \infty$, and has an approximately linear region in between. The regions of the function that the neural network actually utilizes in its feedforward pass are superimposed as histograms on this plot. Interestingly, at initialization, the network does not use the saturation regimes. The inputs to the function are tightly concentrated around $x = 0$ for all instances of the activation function throughout the network. As training progresses, the network makes use of a larger domain of the activation function, and by the time training has concluded the network uses the saturation regimes at approximately $x < -4$ and $x > 1$.

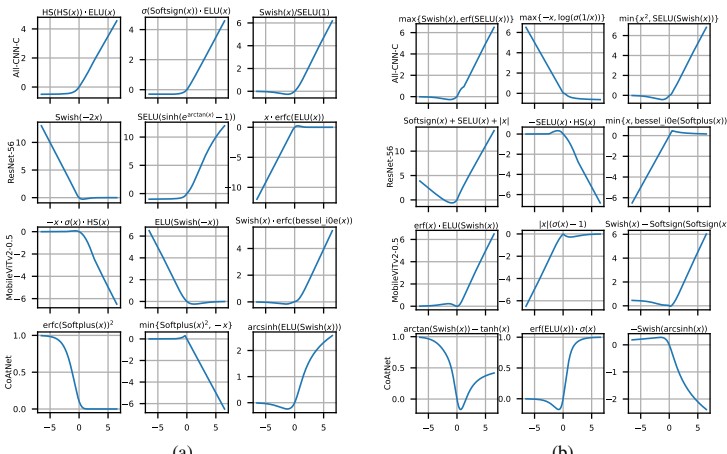

(a)                    (b)

Figure 8: Sample activation functions discovered with AQuaSurF in the four searches in Section 6. "HS" stands for HardSigmoid. (a) The top three functions (columns) discovered in each search (rows). Many of these functions are refined versions of existing activation functions like ELU and Swish. (b) Selected novel activation functions. All of these functions outperformed ReLU and are distinct from existing activation functions. Such designs may serve as a foundation for further improvement and specialization in new settings.

Thus, Figure 9 shows that $\mathrm{erfc}(\mathrm{Softplus}(x))^2$ serves a dual purpose. At initialization, it performs like a rectifier nonlinearity, but by the end of training, it acts like a sigmoidal nonlinearity. This discovery challenges conventional wisdom about activation function design. It shows that neural networks use activation functions in different ways in the different stages of training, and suggests that sigmoidal designs may play an important role after all.

# 8 Conclusion

This paper introduced three benchmark datasets, `Act-Bench-CNN`, `Act-Bench-ResNet`, and `Act-Bench-ViT`, to support research on activation function optimization. Experiments with these datasets showed that FIM eigenvalues and activation function outputs, and their low-dimensional UMAP embeddings, predict activation function performance accurately, and can thus be used as a surrogate for finding better functions, even with out-of-the-box regression algorithms. These conclusions extended from the benchmark datasets to challenging real-world tasks, where better functions were discovered with a variety of datasets, search spaces, and architectures. AQuaSurF also discovered a highly performant sigmoidal activation function,

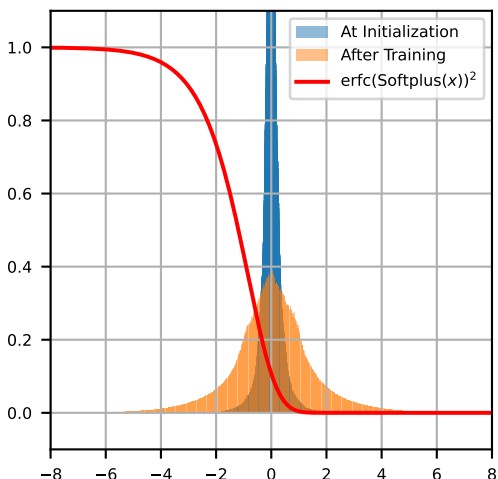

Figure 9: The best discovered function in the CoAt-Net experiment, $\mathrm{erfc}(\mathrm{Softplus}(x))^2$, and its utilization by the network. The red curve shows the activation function itself, and the two histograms show the distributions of inputs to the activation function at initialization and after training, aggregated across all instances of the activation function in the entire network. The network uses the function like a rectifier at initialization and like a sigmoidal activation function after training. This result suggests that sigmoidal designs may be powerful after all, thus challenging the conventional wisdom.

challenging the conventional wisdom of using ReLU-like functions exclusively in deep learning. The study reinforces the idea that activation function design is an important part of deep learning, and shows AQuaSurF is an efficient and flexible mechanism for doing it.

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

## A  Related Work

The techniques in this paper were inspired by prior research in multiple areas, including neural architecture and activation function search, as well as research on the FIM.

**Neural Architecture Search**   In neural architecture search [NAS; 13, 57, 60], the goal is to design a neural network architecture automatically. NAS approaches typically focus on optimizing the type and location of the layers and the connections between them, but often use standard activation functions like ReLU. This work in complimentary to NAS approaches, because it uses standard architectures but optimizes the design of the activation function.

**Zero-cost NAS Proxies**   Recently, zero-cost NAS proxies have received increased attention [42, 50, 56]. These approaches aim to accelerate neural architecture search by using cheap surrogate calculations in place of expensive full training of architectures. This paper adopts a similar approach, using FIM eigenvalues and activation function outputs to predict which activation functions are likely to be most promising before dedicating resources to evaluating them.

**Activation Function Search**   Methods for automatically discovering activation functions include reinforcement learning [49], evolutionary computation [3, 5, 6, 35], and gradient-based methods [1, 5, 19, 44, 54]. This paper builds upon existing work, focusing on efficient search and on understanding the properties that make activation functions effective.

**Other Uses of the FIM**   This paper used FIM eigenvalues to predict the performance of different activation functions. The FIM is an important quantity in machine learning with several uses. One important example is optimal experiment design [14], where experiments are designed to be optimal with respect some criterion. The criteria vary, but are often functions of the eigenvalues of the FIM, such as the maximum or minimum eigenvalue, or the trace of the FIM (sum of the eigenvalues) or determinant of the FIM (product of the eigenvalues). Instead of choosing one optimality criterion and only considering one summary statistic, this paper keeps all of the eigenvalues of the FIM and learns an optimal distribution experimentally.

Past work has also used the eigenvalues of the FIM to determine suitable values of the batch size or learning rate for neural networks [15, 16, 21, 29, 34]. The FIM provides insights to the learning dynamics of SGD [27] and the dynamics of signal propagation at different layers in networks with and without batch normalization layers [25]. The FIM has also been used to develop second-order optimization algorithms for neural networks [20, 38, 39]. Applying it to activation function design is thus a compelling further opportunity.

## B  Activation Function Search Spaces

The activation functions in this paper were implemented as computation graphs from the PANGAEA search space [5]. The space includes unary and binary operators, in addition to existing activation functions [7, 12, 30, 45, 49]. This approach allows specifying families of functions in a compact manner. It is thus possible to focus the search on a space where good functions are likely to be located, and also to search it comprehensively.

**Benchmark Datasets**   The benchmark datasets introduced in Section 2 contain every activation function of the three-node form `binary(unary(`$x$`),unary(`$x$`))` using the operators in Table 4. The result is 5,103 activation functions, of which 2,913 are unique. This space is visualized in Figure 4.

For `Act-Bench-CNN` and `Act-Bench-ResNet`, the accuracies are the median from three runs. For `Act-Bench-ViT`, the results are from single runs due to computational costs.

**New Settings**   The first experiment in Section 6 utilized a larger search space. Specifically, it was based on the following four-node computation graphs: `binary(unary(unary(`$x$`)),unary(`$x$`))`, `binary(unary(`$x$`),unary(unary(`$x$`)))`,  `n-ary(unary(`$x$`),unary(`$x$`),unary(`$x$`))`, `unary(binary(unary(`$x$`),unary(`$x$`)))`, and `unary(unary(unary(unary(`$x$`))))`.   The unary and binary nodes used the operators in Table 4, and the $n$-ary node used the sum, product,

Table 4: Activation function search spaces were defined through computation graphs consisting of basic unary and binary operators as well as existing activation functions [5].

| Unary | | | Binary |
|---|---|---|---|
| $0$ | $\text{erf}(x)$ | $\text{ReLU}(x)$ | $x_1 + x_2$ |
| $1$ | $\text{erfc}(x)$ | $\text{ELU}(x)$ | $x_1 - x_2$ |
| $x$ | $\sinh(x)$ | $\text{SELU}(x)$ | $x_1 \cdot x_2$ |
| $-x$ | $\cosh(x)$ | $\text{Swish}(x)$ | $x_1/x_2$ |
| $|x|$ | $\tanh(x)$ | $\text{Softplus}(x)$ | $x_1^{x_2}$ |
| $x^{-1}$ | $\text{arcsinh}(x)$ | $\text{Softsign}(x)$ | $\max\{x_1, x_2\}$ |
| $x^2$ | $\arctan(x)$ | $\text{HardSigmoid}(x)$ | $\min\{x_1, x_2\}$ |
| $e^x$ | $e^x - 1$ | $\text{bessel\_i0e}(x)$ | |
| $\sigma(x)$ | $\log(\sigma(x))$ | $\text{bessel\_i1e}(x)$ | |

maximum, and minimum operators. Together, these computation graphs create a search space with 1,023,516 functions, of which 425,896 are unique. This space is visualized in Figures 10 and 11.

The third experiment in Section 6, i.e. CoAtNet on Imagenette, added ELiSH, GELU, HardSigmoid, Leaky ReLU, and Mish as `unary` operators to the original benchmark search space. In this experiment, the search space comprised functions of the form `binary(unary(unary(x)),unary(x))` and `unary(unary(unary(x)))`. This search space contains 238,341 activation functions, of which 146,779 are unique.

## C   Fisher Information Matrix Details

In order to calculate the FIM, this paper uses the K-FAC approach [20, 38, 39]. This technique is summarized in this Appendix, with notation similar to that of Grosse and Martens [20].

**Preliminaries**   A feedforward neural network maps an input $\mathbf{a}_0 = \mathbf{x}$ to an output $\mathbf{a}_L = f(\mathbf{x}; \boldsymbol{\theta})$ through a series of $L$ layers. Each layer $l \in \{1, \ldots, L\}$ is comprised of a weight matrix $\mathbf{W}_l$, a bias vector $\mathbf{b}_l$, and an element-wise activation function $\phi_l$. With $\bar{\mathbf{W}}_l = (\mathbf{b}_l \quad \mathbf{W}_l)$ and $\bar{\mathbf{a}}_l = \begin{pmatrix} 1 & \mathbf{a}_l^\top \end{pmatrix}^\top$, each layer implements the transformation

$$\mathbf{s}_l = \bar{\mathbf{W}}_l \bar{\mathbf{a}}_{l-1}, \tag{4}$$

$$\mathbf{a}_l = \phi_l(\mathbf{s}_l). \tag{5}$$

Let $\boldsymbol{\theta} = \begin{pmatrix} \text{vec}(\bar{\mathbf{W}}_1)^\top & \cdots & \text{vec}(\bar{\mathbf{W}}_L)^\top \end{pmatrix}^\top$ represent the vector of all network parameters. Parameterized by $\boldsymbol{\theta}$ and given inputs $\mathbf{x}$ drawn from a training distribution $Q_\mathbf{x}$, the neural network defines the conditional distribution $R_{\mathbf{y}|f(\mathbf{x};\boldsymbol{\theta})}$. The Fisher information matrix associated with this model is

$$\mathbf{F} = \mathop{\mathbb{E}}_{\substack{\mathbf{x} \sim Q_\mathbf{x} \\ \mathbf{y} \sim R_{\mathbf{y}|f(\mathbf{x};\boldsymbol{\theta})}}} \left[ \nabla_{\boldsymbol{\theta}} \mathcal{L}(\mathbf{y}, f(\mathbf{x}; \boldsymbol{\theta})) \nabla_{\boldsymbol{\theta}} \mathcal{L}(\mathbf{y}, f(\mathbf{x}; \boldsymbol{\theta}))^\top \right]. \tag{6}$$

As usual in deep learning, the loss function $\mathcal{L}(\mathbf{y}, \mathbf{z})$ represents the negative log-likelihood associated with $R_{\mathbf{y}|f(\mathbf{x};\boldsymbol{\theta})}$ and quantifies the discrepancy between the model's prediction $\mathbf{z} = f(\mathbf{x}; \boldsymbol{\theta})$ and the true label $\mathbf{y}$. The network is trained to minimize the loss by updating its parameters according to the gradient $\nabla_{\boldsymbol{\theta}} \mathcal{L}(\mathbf{y}, f(\mathbf{x}; \boldsymbol{\theta}))$.

**Approximations**   For ease of notation, write $\mathcal{D}\mathbf{v} = \nabla_\mathbf{v} \mathcal{L}(\mathbf{y}, f(\mathbf{x}; \boldsymbol{\theta}))$. Recalling that $\boldsymbol{\theta} = \begin{pmatrix} \text{vec}(\bar{\mathbf{W}}_1)^\top & \cdots & \text{vec}(\bar{\mathbf{W}}_L)^\top \end{pmatrix}^\top$, the FIM can be expressed as an $L \times L$ block matrix:

$$\mathbf{F} = \begin{pmatrix} \mathbb{E}\left[\text{vec}(\mathcal{D}\bar{\mathbf{W}}_1)\text{vec}(\mathcal{D}\bar{\mathbf{W}}_1)^\top\right] & \cdots & \mathbb{E}\left[\text{vec}(\mathcal{D}\bar{\mathbf{W}}_1)\text{vec}(\mathcal{D}\bar{\mathbf{W}}_L)^\top\right] \\ \vdots & \ddots & \vdots \\ \mathbb{E}\left[\text{vec}(\mathcal{D}\bar{\mathbf{W}}_L)\text{vec}(\mathcal{D}\bar{\mathbf{W}}_1)^\top\right] & \cdots & \mathbb{E}\left[\text{vec}(\mathcal{D}\bar{\mathbf{W}}_L)\text{vec}(\mathcal{D}\bar{\mathbf{W}}_L)^\top\right] \end{pmatrix}. \tag{7}$$

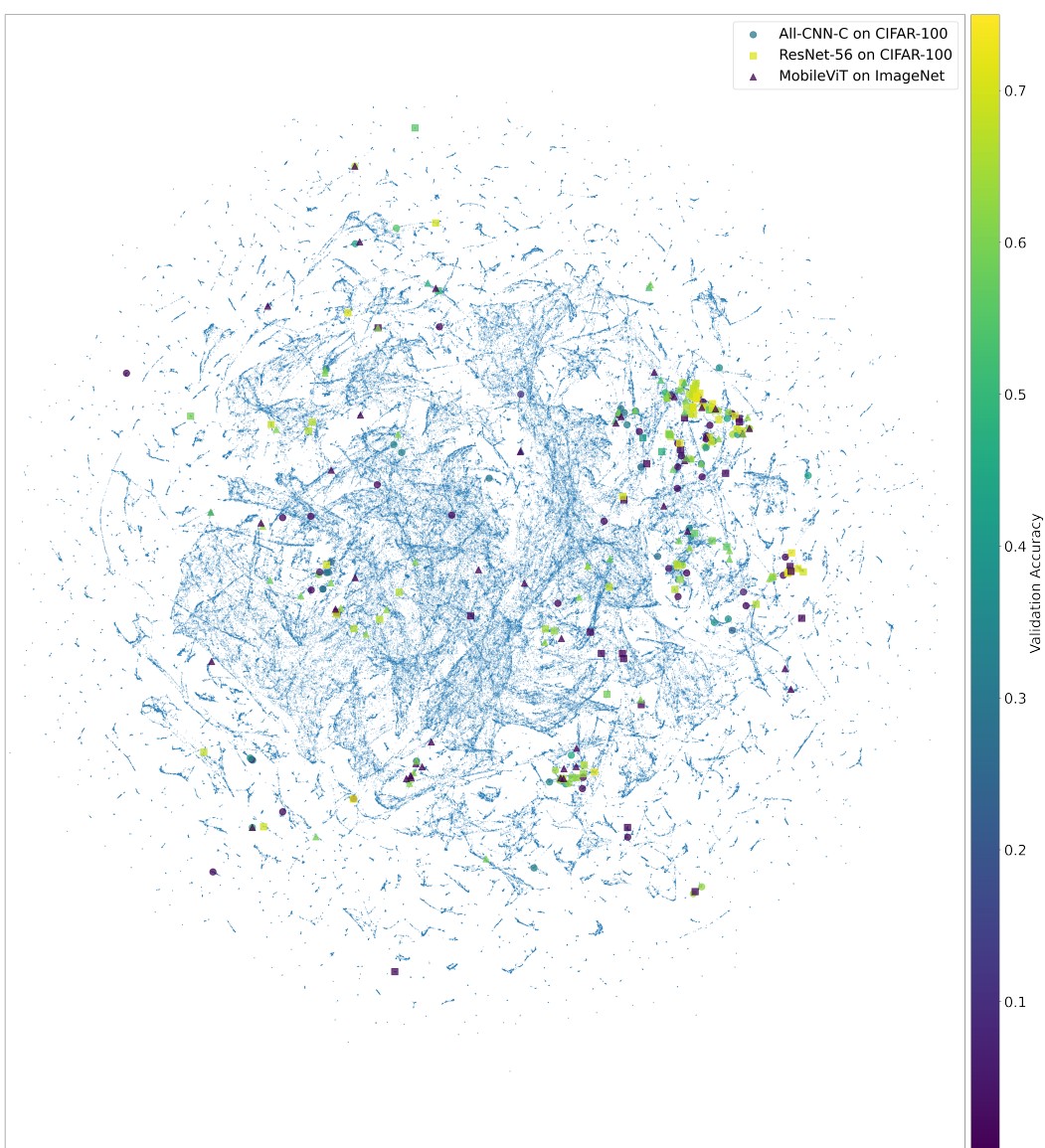

Figure 10: Low-dimensional UMAP representation of the 425,896 function search space. The activation functions are embedded according to their outputs; each point represents a unique function. The larger points represent activation functions that were evaluated during the searches; they are colored according to their validation accuracy. Although the space is vast, the searches require only tens of evaluations to discover good activation functions.

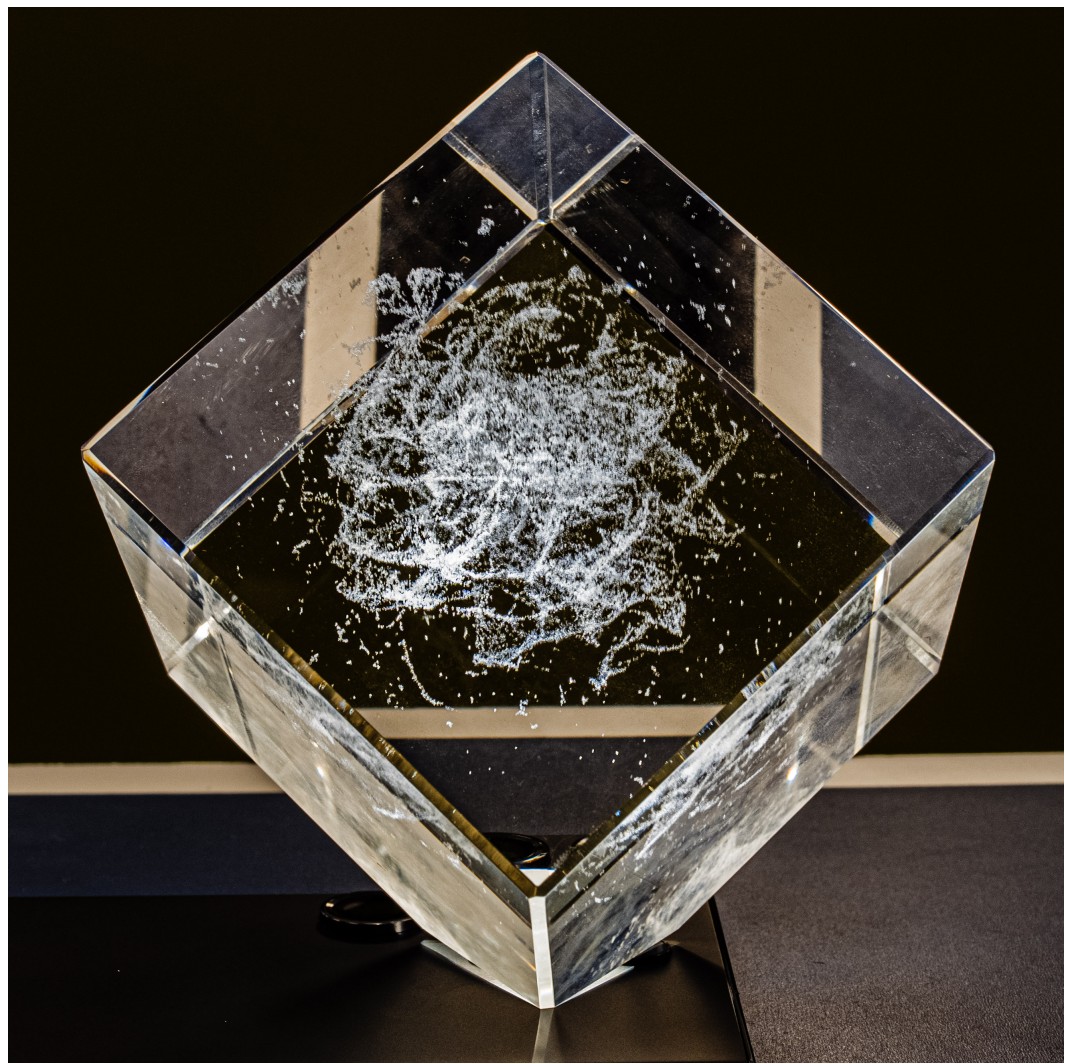

Figure 11: A photograph of a three-dimensional scatter plot laser-engraved into a physical crystal cube. Each point represents one of the unique 425,896 unique activation functions in the search space. Points are arranged according to a 3D UMAP projection according to activation function outputs; the points are the same as those shown in Figure 10. The cube shows the size and complexity of the search space, and the 1D and 2D manifolds reveal the underlying structure.

Note that $\mathcal{D}\bar{\mathbf{W}}_l = \mathcal{D}\mathbf{s}_l\bar{\mathbf{a}}_{l-1}^\top$, and recall that $\text{vec}(\mathbf{u}\mathbf{v}^\top) = \mathbf{v} \otimes \mathbf{u}$. Each block of the FIM can be written as

$$\mathbf{F}_{i,j} = \mathbb{E}\left[\text{vec}(\mathcal{D}\bar{\mathbf{W}}_i)\text{vec}(\mathcal{D}\bar{\mathbf{W}}_j)^\top\right] \tag{8}$$

$$= \mathbb{E}\left[\text{vec}(\mathcal{D}\mathbf{s}_i\bar{\mathbf{a}}_{i-1}^\top)\text{vec}(\mathcal{D}\mathbf{s}_j\bar{\mathbf{a}}_{j-1}^\top)^\top\right] \tag{9}$$

$$= \mathbb{E}\left[(\bar{\mathbf{a}}_{i-1} \otimes \mathcal{D}\mathbf{s}_i)(\bar{\mathbf{a}}_{j-1} \otimes \mathcal{D}\mathbf{s}_j)^\top\right] \tag{10}$$

$$= \mathbb{E}\left[(\bar{\mathbf{a}}_{i-1} \otimes \mathcal{D}\mathbf{s}_i)(\bar{\mathbf{a}}_{j-1}^\top \otimes \mathcal{D}\mathbf{s}_j^\top)\right] \tag{11}$$

$$= \mathbb{E}\left[\bar{\mathbf{a}}_{i-1}\bar{\mathbf{a}}_{j-1}^\top \otimes \mathcal{D}\mathbf{s}_i\mathcal{D}\mathbf{s}_j^\top\right]. \tag{12}$$

Two approximations are necessary in order to make representation of the FIM practical. First, assume that different layers have uncorrelated weight derivatives. The FIM can then be approximated as a block diagonal matrix, with $\mathbf{F}_{i,j} = \mathbf{0}$ if $i \neq j$. Second, if one approximates the pre-activation derivatives $\mathcal{D}\mathbf{s}_l$ and activations $\bar{\mathbf{a}}_{l-1}^\top$ as independent, then the diagonal blocks of the FIM can be

further decomposed into the Kronecker product of two smaller matrices:

$$\mathbf{F}_{l,l} = \mathbb{E}\left[\bar{\mathbf{a}}_{l-1}\bar{\mathbf{a}}_{l-1}^\top \otimes \mathcal{D}\mathbf{s}_l\mathcal{D}\mathbf{s}_l^\top\right] \approx \mathbb{E}\left[\bar{\mathbf{a}}_{l-1}\bar{\mathbf{a}}_{l-1}^\top\right] \otimes \mathbb{E}\left[\mathcal{D}\mathbf{s}_l\mathcal{D}\mathbf{s}_l^\top\right]. \tag{13}$$

Let $\mathbf{\Omega}_l = \mathbb{E}\left[\bar{\mathbf{a}}_l\bar{\mathbf{a}}_l^\top\right]$ and $\mathbf{\Gamma}_l = \mathbb{E}\left[\mathcal{D}\mathbf{s}_l\mathcal{D}\mathbf{s}_l^\top\right]$. The approximate empirical FIM is then written as

$$\hat{\mathbf{F}} = \begin{pmatrix} \mathbf{\Omega}_0 \otimes \mathbf{\Gamma}_1 & & \mathbf{0} \\ & \ddots & \\ \mathbf{0} & & \mathbf{\Omega}_{L-1} \otimes \mathbf{\Gamma}_L \end{pmatrix}. \tag{14}$$

**Layer-Specific Implementation**   The above example illustrates FIM approximation for a simple feedforward network. However, most modern architectures contain several different kinds of layers. Some layers like pooling, normalization, or dropout layers do not have trainable weights, and therefore these layers are not included in the FIM [26, 52].

Each diagonal entry $\mathbf{\Omega}_{l-1} \otimes \mathbf{\Gamma}_l$ corresponds to one layer with weights. The calculation differs slightly depending on the layer type, but otherwise the example above can be straightforwardly extended to more complicated networks. Calculations for three common layer types are presented below.

**Dense Layers**   For dense layers, the matrices $\mathbf{\Omega}_{l-1}$ and $\mathbf{\Gamma}_l$ can be readily computed with one forward and backward pass through the network using a mini-batch of data. The eigenvalues are then computed using standard techniques.

**Convolutional Layers**   Convolutional layers require special consideration to calculate $\mathbf{\Omega}_{l-1}$ and $\mathbf{\Gamma}_l$. For a given layer, let $M$ represent the batch size, $\mathcal{T}$ the set of spatial locations (typically a two-dimensional grid), $\Delta$ the set of spatial offsets from the center of the filter, and $I$ and $J$ the number of output and input maps, respectively. The activations are represented by the $M \times |\mathcal{T}| \times J$ array $\mathbf{A}_{l-1}$. The weights are represented by the $I \times |\Delta| \times J$ array $\mathbf{W}_l$ which is interpreted as an $I \times |\Delta|J$ matrix. The expansion operator $[\![\cdot]\!]$ extracts patches around each spatial location and flattens them into vectors that become the rows of a matrix: $[\![\mathbf{A}_{l-1}]\!]$ is a $M|\mathcal{T}| \times J|\Delta|$ matrix.

Similar to the feedforward networks, the bias (if used) can be prepended to the weights matrix as $\bar{\mathbf{W}}_l = (\mathbf{b}_l \quad \mathbf{W}_l)$ and a homogeneous column of ones to the expanded activations as $[\![\mathbf{A}_{l-1}]\!]_H = (\mathbf{1} \quad [\![\mathbf{A}_{l-1}]\!])$. This constructions allows the forward pass to be written as

$$\mathbf{S}_l = [\![\mathbf{A}_{l-1}]\!]_H \bar{\mathbf{W}}_l^\top, \tag{15}$$
$$\mathbf{A}_l = \phi\left(\mathbf{S}_l\right), \tag{16}$$

from which the factors are computed as

$$\mathbf{\Omega}_l = \mathbb{E}\left[[\![\mathbf{A}_l]\!]_H^\top[\![\mathbf{A}_l]\!]_H\right], \tag{17}$$
$$\mathbf{\Gamma}_l = \frac{1}{|\mathcal{T}|}\mathbb{E}\left[\mathcal{D}\mathbf{S}_l^\top\mathcal{D}\mathbf{S}_l\right]. \tag{18}$$

**Depthwise Convolutional Layers**   Depthwise convolutional layers utilize separate kernels for each channel. In this case, $[\![\mathbf{A}_{l-1}]\!]$ is a $M|\mathcal{T}|J \times |\Delta|$ matrix. Otherwise, the factors $\mathbf{\Omega}_{l-1}$ and $\mathbf{\Gamma}_l$ are calculated in the same way as they are for standard convolutional layers.

**Eigenvalue Calculation**   Because $\hat{\mathbf{F}}$ is a block-diagonal matrix, its eigenvalues are simply the combined eigenvalues of each block: $\lambda(\hat{\mathbf{F}}) = \{\lambda(\hat{\mathbf{F}}_l)\}_{l=1}^L$. The eigenvalue calculation for one block $\hat{\mathbf{F}}_l = \mathbf{\Omega}_{l-1} \otimes \mathbf{\Gamma}_l$ is further simplified by first computing the eigenvalues $\lambda(\mathbf{\Omega}_{l-1})$ and $\lambda(\mathbf{\Gamma}_l)$ for each Kronecker factor separately and then returning all pairwise products from the two sets of eigenvalues. For numerical stability, the eigenvalues can first be log-scaled and then all pairwise sums from the two sets are returned. Calculating the eigenvalues requires one forward and backward pass through the network with a mini-batch of data. The computational cost is therefore relatively cheap, especially compared with the cost of fully training a network from scratch.

It is possible for the FIM eigenvalues to be invalid. For example, if the forward propagated activations or backward propagated gradients explode or vanish, then the diagonal entries $\mathbf{\Omega}_{l-1} \otimes \mathbf{\Gamma}_l$ may be undefined. Such invalid values result from activation functions that are unstable. Therefore, invalid FIM eigenvalues provide a good way to filter out bad activation functions.

# D   Features and Surrogate Details

This section describes how the activation function features were implemented and how the surrogate was constructed.

**Calculating FIM Eigenvalues**   The FIM eigenvalues were calculated for each activation function as discussed in Section 3. The eigenvalues were log-scaled for numerical stability. By definition, the number of eigenvalues is the same as the number of weights in the neural network. To save space, the eigenvalues were binned to histograms. For a layer $l$ with $|\boldsymbol{\theta}_l|$ weights, $\lfloor|\boldsymbol{\theta}_l|/100\rfloor$ equally sized bins from $-100$ to $100$ were used. One histogram was computed for each layer in a network, and all of the histograms were concatenated together into a single feature vector for a given activation function. In this manner, the total dimensionality was 13,692 for All-CNN-C, 16,500 for ResNet-56, and 11,013 for MobileViTv2-0.5.

**Calculating Activation Function Outputs**   The activation function outputs $y = f(x)$ were calculated for each activation function $f$ by sampling $n =$1,000 values $x \sim \mathcal{N}(0, 1)$ and truncating to the range $[-5, 5]$. The same random inputs were used for all activation functions.

**Per-Layer FIM Eigenvalues**   In Figure 5, the eigenvalues for the entire network are shown for completeness. However, the UMAP representations shown in Figure 4 were produced by keeping the eigenvalues at each layer separate and computing a weighted distance between them (according to Equation 2). As pointed out in the main text, FIM eigenvalues are informative but noisy features. In preliminary experiments, keeping the eigenvalues separate at each layer reduced some of this noise, resulting in a more informative Figure 4 and consequently improving the performance of the search algorithms.

**FIM Eigenvalue Features**   Preliminary experiments aimed to predict activation function performance using common features in the literature, including maximum eigenvalue, minimum eigenvalue, sum of the eigenvalues, and product of the eigenvalues [14]. More recently proposed features, such as (second moment) / (first moment)$^2$, were also considered [48]. Ultimately, learning the relevant features from the entire eigenvalue distribution was found to be the most flexible and powerful approach.

**UMAP Settings**   UMAP exposes a number of parameters that can be used to customize its behavior [40]. The `metric` parameter determines how distances are computed between points, the `n_neighbors` parameter adjusts the tradeoff between the local and global structure of the data, and the `min_dist` parameter controls the minimum distance between points in the embedding space.

The plots in Figure 4 were produced by computing the distances between FIM eigenvalues and activation function outputs. For the FIM eigenvalues `UMAP(metric='manhattan', n_neighbors=3, min_dist=0.1)` was used, and for the activation function outputs `UMAP(metric='euclidean', n_neighbors=15, min_dist=0.1)` was used. The distance metrics were chosen to implement Equations 2 and 3.

In preliminary experiments, decreasing `n_neighbors` from the default of 15 down to 3 for the FIM eigenvalues qualitatively improved the embedding for the combined features. The combined features were visualized with a union model, i.e. `umap_combined = umap_fim_eigs + umap_fn_outputs` [40].

# E   Experiment Details

This section specifies the details for the experiments in the main text of the paper. Several variations to the approach presented in the main text were also evaluated in preliminary experiments. The approach turned out to be robust to most of them, but the results also justify the choices used for the main experiments.

**Training Details**   For CIFAR-10 and CIFAR-100, balanced validation sets were created by sampling 5,000 images from the training set. Full training details and hyperparameters are listed in Tables 5 and 6.

Table 5: Training details and hyperparameter values used in the CIFAR-10 and CIFAR-100 experiments.

| All-CNN-C on CIFAR-10 and CIFAR-100 | |
|---|---|
| Batch Size | 128 |
| Dropout | 0.5 |
| Epochs | 25 for `Act-Bench-CNN` and search (Figure 7), 50 for full evaluation (Table 1) |
| Image Size | $32 \times 32$ |
| Learning Rate | Linear warmup to 0.1 for five epochs, then linear decay |
| Mean/Std. Normalization | Yes |
| Momentum | 0.9 |
| Optimizer | SGD |
| Random Crops | $32 \times 32$ crops of images padded with four pixels on all sides |
| Random Flips | Yes |
| Weight Decay | $1e^{-4}$ |
| Weight Initialization | AutoInit [4] |

| ResNet-56 on CIFAR-10 and CIFAR-100 | |
|---|---|
| Batch Size | 128 |
| Dropout | 0.0 |
| Epochs | 25 for `Act-Bench-ResNet` and search (Figure 7), 50 for full evaluation (Table 1) |
| Image Size | $32 \times 32$ |
| Learning Rate | Linear warmup to 0.1 for five epochs, then linear decay |
| Mean/Std. Normalization | No |
| Momentum | 0.9 |
| Optimizer | SGD |
| Random Crops | $32 \times 32$ crops of images padded with five pixels on all sides |
| Random Flips | Yes |
| Weight Decay | $1e^{-4}$ |
| Weight Initialization | AutoInit [4] |

**CoAtNet** A smaller variant of the CoAtNet architecture[2] was used in order to fit the model and data on the available GPU memory. The architecture has three convolutional blocks with 64 channels, four convolutional blocks with 128 channels, six transformer blocks with 256 channels, and three transformer blocks with 512 channels. This architecture is slightly deeper but thinner than the original CoAtNet-0 architecture, which has two convolutional blocks with 96 channels, three convolutional blocks with 192 channels, five transformer blocks with 384 channels, and two transformer blocks with 768 channels [10]. The models are otherwise identical.

**Search Implementation** In order to predict performance for an unevaluated activation function, the function outputs and FIM eigenvalues must first be computed. Thus, the searches in Section 6 were implemented in three steps. First, activation function outputs for all 425,896 activation functions in the search space were calculated. This computation is inexpensive and easily parallelizable. Second, eight workers operated in parallel to sample activation functions uniformly at random from the search space and calculate their FIM eigenvalues. Third, once the number of activation functions with FIM eigenvalues calculated reached 5,000, seven of the workers began the search by evaluating the functions with the highest predicted performance. The eighth worker continued calculating FIM eigenvalues for new functions so that their performance could be predicted during the search. This setup allowed taking best advantage of the available compute for the regression-type search methods.

The experiments on ImageNet required substantially more compute than the experiments on CIFAR-100. For this reason, all eight workers evaluated activation functions once the number of functions with FIM eigenvalues reached 7,000.

Computing FIM eigenvalues took approximately 26 seconds, 84 seconds, and 37 seconds per activation function for All-CNN-C, ResNet-56, and MobileViTv2-0.5, respectively. This cost is not trivial, but it is well worth it, as the experiments in the main paper show.

---

[2]`https://github.com/leondgarse/keras_cv_attention_models/blob/v1.3.0/keras_cv_attention_models/coatnet/coatnet.py#L199`

Table 6: Training details and hyperparameter values used in the Imagenette and ImageNet experiments.

| MobileViTv2-0.5 on Imagenette and ImageNet | |
|---|---|
| Batch Size | 256 |
| CutMix Alpha [58] | 1.0 |
| Epochs | 105 |
| Evaluation Center Crop | 95% |
| Image Size | $160 \times 160$ |
| Learning Rate | Linear warmup from $1e^{-4}$ to $4e^{-3}$ for five epochs, then cosine decay to $1e^{-6}$ |
| Mixup Alpha [59] | 0.1 |
| Optimizer | AdamW [36] |
| RandAugment [9] | Magnitude six, applied twice |
| Random Resized Crop [53] | Minimum 8% of the original image |
| Weight Decay | $0.02\times$ current learning rate |

| ResNet-50 on ImageNet | |
|---|---|
| Batch Size | 256 |
| CutMix Alpha [58] | 1.0 |
| Epochs | 105 |
| Evaluation Center Crop | 95% |
| Image Size | $160 \times 160$ |
| Learning Rate | Linear warmup from $1e^{-4}$ to $2e^{-3}$ for five epochs, then cosine decay to $1e^{-6}$ |
| Mixup Alpha [59] | 0.1 |
| Optimizer | AdamW [36] |
| RandAugment [9] | Magnitude six, applied twice |
| Random Resized Crop [53] | Minimum 8% of the original image |
| Weight Decay | $0.02\times$ current learning rate |
| Weight Initialization | AutoInit [4] |

| CoAtNet on Imagenette | |
|---|---|
| Batch Size | 256 |
| CutMix Alpha [58] | 1.0 |
| Epochs | 105 |
| Evaluation Center Crop | 95% |
| Image Size | $160 \times 160$ |
| Learning Rate | Linear warmup from $1e^{-4}$ to $4e^{-4}$ for five epochs, then cosine decay to $1.6e^{-7}$ |
| Mixup Alpha [59] | 0.1 |
| Optimizer | AdamW [36] |
| RandAugment [9] | Magnitude six, applied twice |
| Random Resized Crop [53] | Minimum 8% of the original image |
| Weight Decay | $0.02\times$ current learning rate |

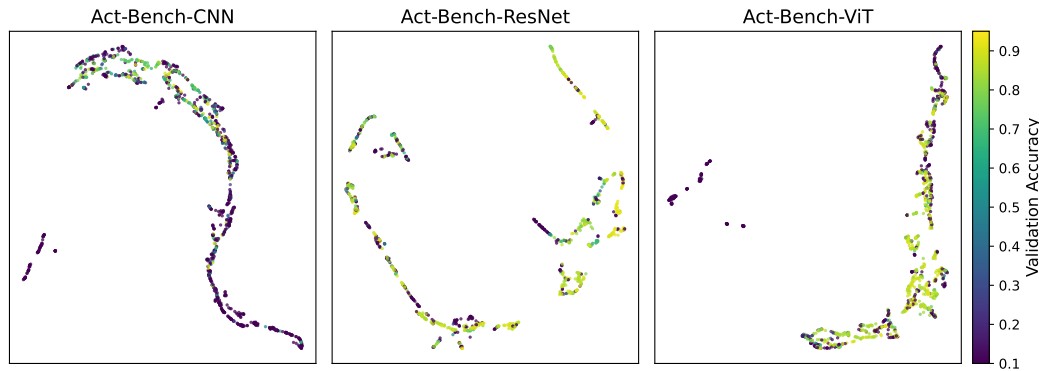

Figure 12: UMAP projections of FIM eigenvalues using the default hyperparameter of `n_neighbors=15`. The embedding is informative but also noisy. Using `n_neighbors=3`, as shown in the main text, improved performance.

**Unique Activation Functions**    Different computation graphs can result in the same activation function (e.g. $\max\{x, 0\}$ and $\max\{0, x\}$). In the benchmark dataset and in the larger search space of Section 6, repeated activation functions were filtered out. 1,000 inputs were sampled $\mathcal{N}(0, 1)$ and truncated to $[-5, 5]$. Two activation functions were considered the same if their outputs were identical.

**Improving the Combined UMAP Projection**    Figure 12 displays a projection of FIM eigenvalues using default UMAP hyperparameters. The plots show the eigenvalues organized in multiple distinct one-dimensional manifolds. Again, FIM eigenvalues are noisy features; there are some clusters of activation functions achieving similar performance, but there are also regions where performance varies widely. As mentioned in the main text, this issue was addressed by reducing the UMAP parameter `n_neighbors` to 3. This change reduced the connectivity of the low-dimensional FIM eigenvalue representation, resulting in a space with many distinct clusters (as seen in Figure 4).

On its own, this setting did not improve the search on the benchmark datasets. However, it did improve performance when the FIM eigenvalues were combined with activation function outputs (as was discussed in Section 4). The reason is that the UMAP model for the activation function outputs did not decrease `n_neighbors`, and so the combined UMAP model relied more on the activation function outputs than it did on the FIM eigenvalues. As Figure 4 shows, the activation function outputs are reliable but sometimes project good activation functions to distinct regions in the search space. Introducing extra connectivity into the fuzzy topological representation via the FIM eigenvalues was sufficient to address this issue, bringing good activation functions to common regions of the space.

**Increasing the Dimension of the UMAP Projections**    The UMAP plots show two-dimensional projections of FIM eigenvalues and activation function outputs. Regression algorithms were also trained on five and 10-dimensional projections. These runs resulted in comparable or worse performance. Therefore, the two-dimensional projections were selected in the paper for simplicity and for consistency between the algorithm implementation and figure visualizations.

**Gaussian Process Regression**    As an alternative search method, Gaussian process regression (GPR) was evaluated in activation function search. Several different acquisition mechanisms were used, including expected improvement, probability of improvement, maximum predicted value, and upper confidence bound. The approach worked well, but the results were inconsistent across the different acquisition mechanisms. GPR was also more expensive to run compared to the algorithms in the main text (KNR, RFR, SVR), and so those algorithms were used instead for simplicity and efficiency.

**Adjusting $k$ in KNR**    The initial experiments with the KNR algorithm used $k = 3$. Experimenting with $k = \{1, 5, 8\}$ did not reliably improve performance, so $k = 3$ was kept.

**Uniformly Spaced Inputs for Activation Function Outputs**    In an alternative implementation, equally spaced inputs from $-5$ to $5$ were given to the activation functions instead of normally distributed inputs. This variation did not noticeably change the quality of the embeddings nor the performance of the search algorithms. Therefore, normal inputs were used for consistency with Equation 3. Figure 3 is the only exception; it used 80 inputs equally spaced from $-5$ to $5$ and increased the UMAP parameter `min_dist` to 0.5. These settings improved the quality of the reconstructed activation functions in the plot.

**Evaluated Functions**    Figures 13, 14, 15, and 16 show plots and the validation accuracy of every candidate activation function evaluated in the searches for All-CNN-C on CIFAR-100, ResNet-56 on CIFAR-100, MobileViTv2-0.5 on ImageNet, and CoAtNet on Imagenette, respectively.

# F   Future Work

This paper demonstrated that FIM eigenvalues and activation function outputs are efficient and reliable features that can predict performance of activation functions accurately. This finding enabled discovering better activation functions for various tasks, improving the state of the art in machine learning. Because the technique is efficient, it was possible to scale it up to large datasets such as ImageNet. These discoveries inspire several avenues for future research, discussed below.

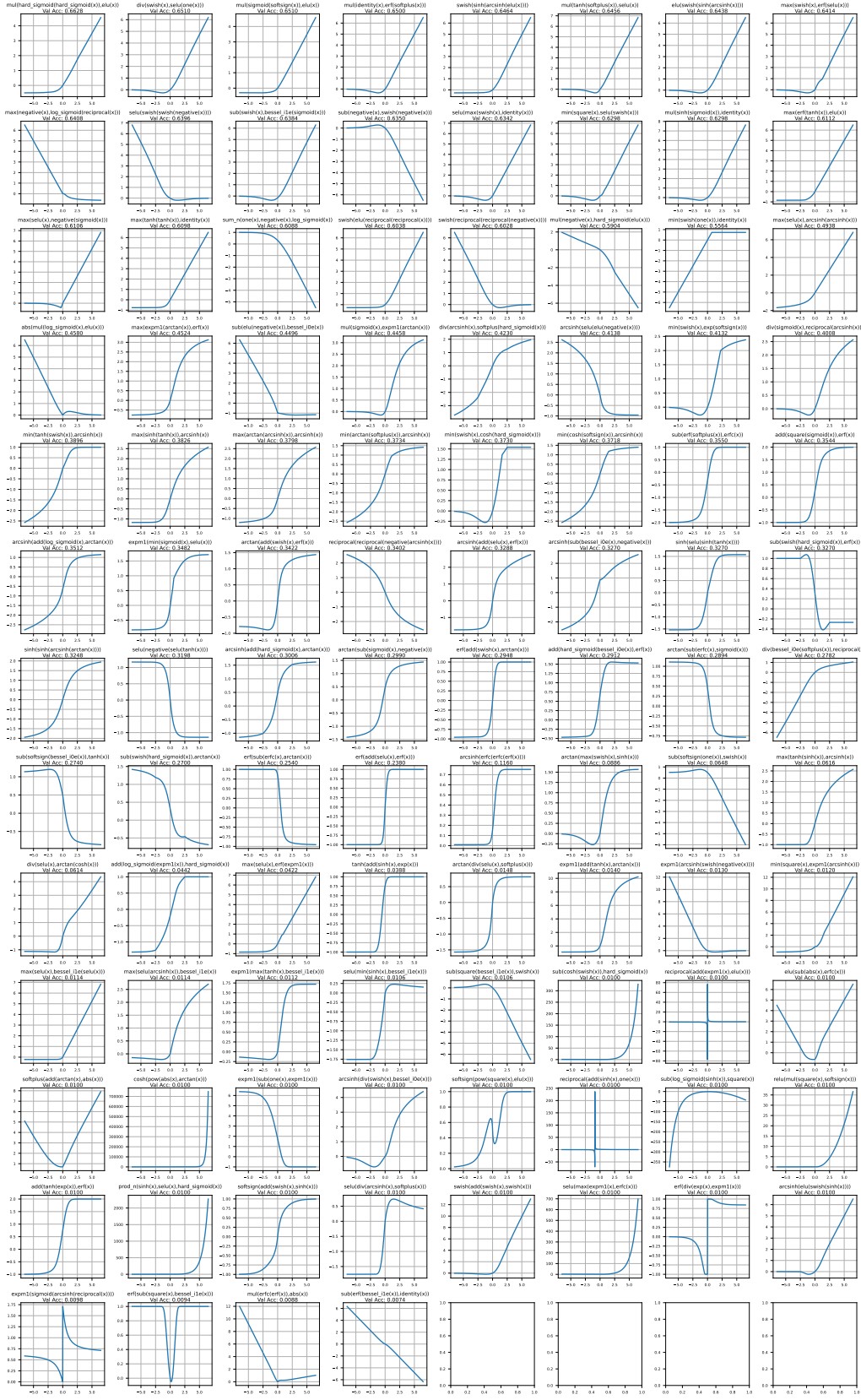

Figure 13: Activation functions evaluated in the search for All-CNN-C on CIFAR-100.

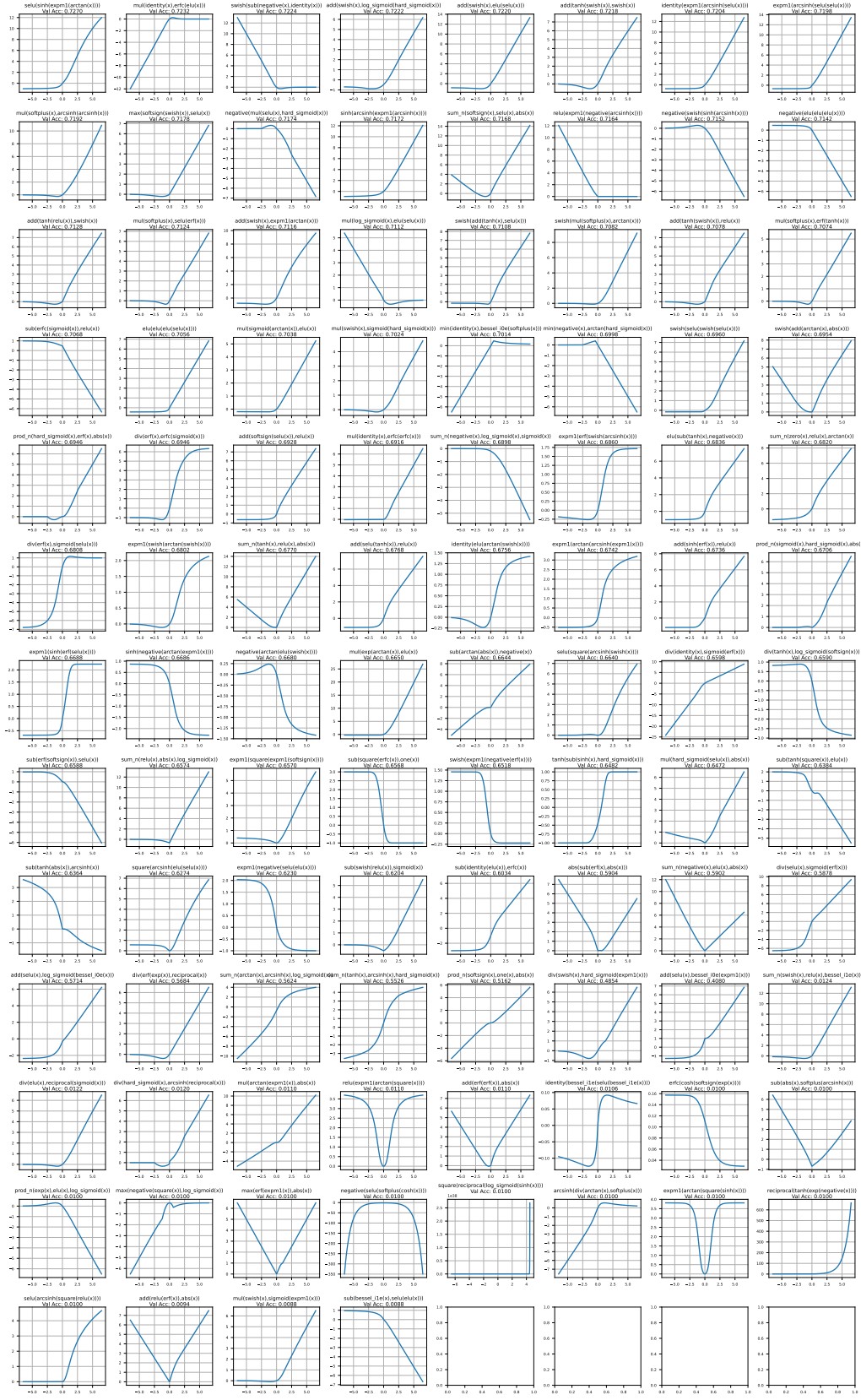

Figure 14: Activation functions evaluated in the search for ResNet-56 on CIFAR-100.

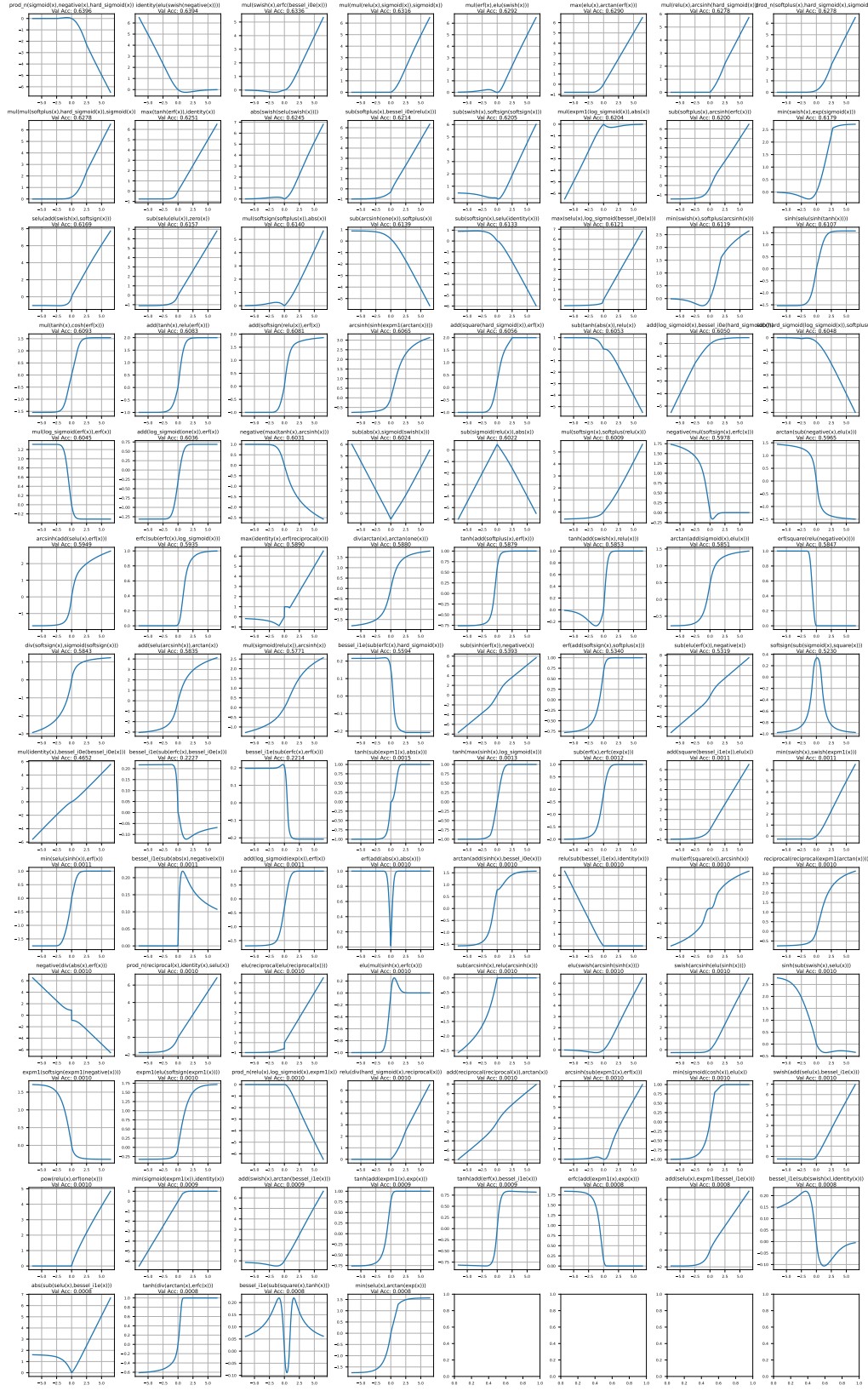

Figure 15: Activation functions evaluated in the search for MobileViTv2-0.5 on ImageNet.

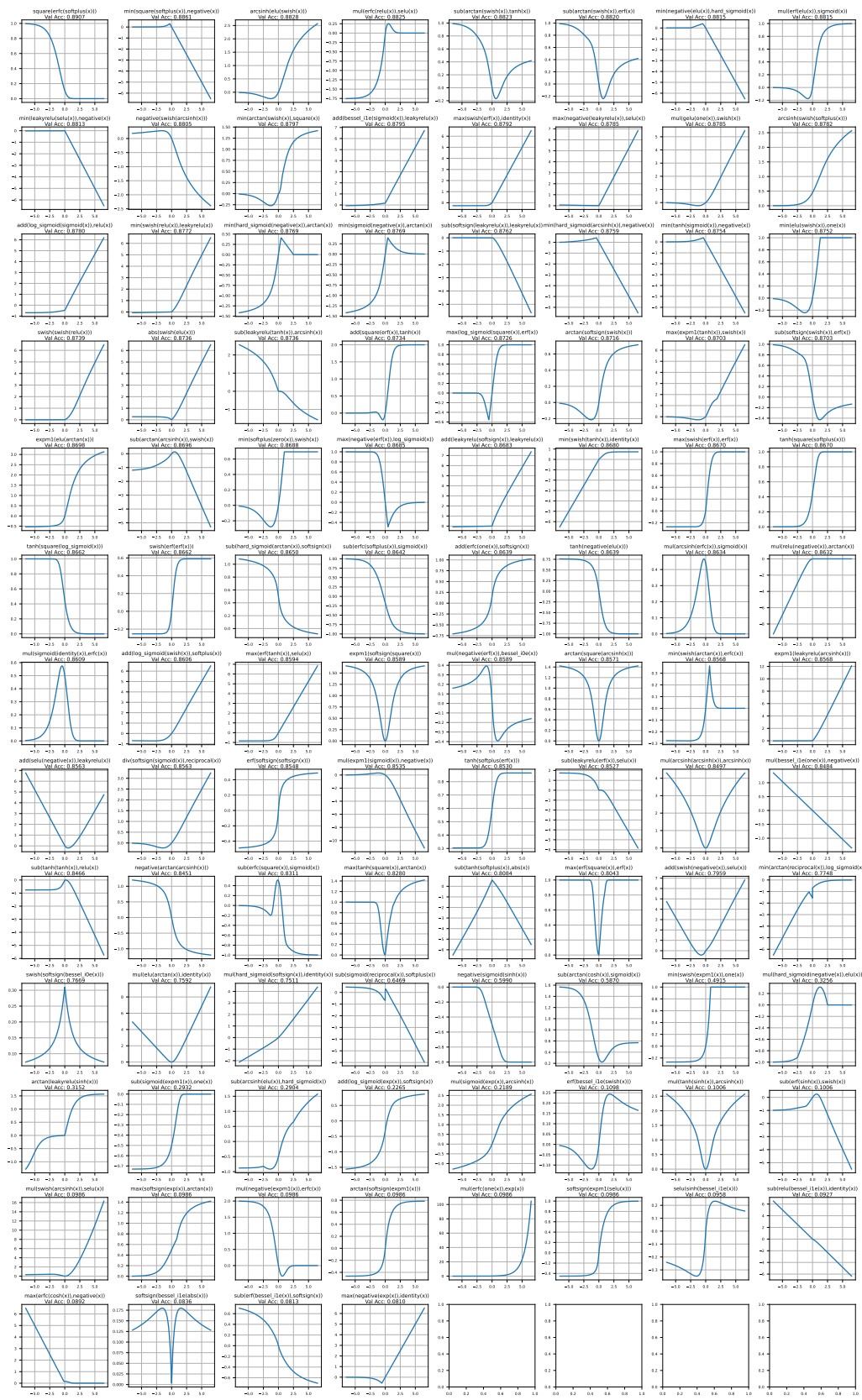

Figure 16: Activation functions evaluated in the search for CoAtNet on Imagenette.

**New Search Spaces**    The PANGAEA search space was used in this paper because it is known to work well for deep architectures [5]. In the future it will be interesting to explore search spaces with different unary, binary, and $n$-ary operators. Beyond computation graphs, it may also be possible to apply techniques in this paper to optimize continuous vector representations of activation functions [1, 44].

**Exploration vs. Exploitation**    The KNR approach was utilized to search for new activation functions because it performed well on the benchmark datasets (Section 5). In the future, it will be interesting to consider other algorithms and analyze their tradeoffs between exploration and exploitation. For example, in a resource-constrained environment where improvement is needed quickly, a more exploitative approach could be used to find an improved activation function in a short time. On the other hand, if substantial compute is available, an approach that focuses on exploration could be used to discover activation functions that perform well but are maximally different from functions used in modern architectures (Figure 8b). Novelty search [32] could serve as a suitable approach, and such discoveries could further understanding of how neural networks utilize different kinds of activation functions to learn.

**Optimizing Multiple Activation Functions**    In a typical neural network design, the same activation function is used throughout the network. However, recent work has shown that it may be beneficial to have different activation functions at different locations, and further, that it may be useful to have different activation functions in the early and late stages of training [5]. Indeed, many hybrid architectures use Swish in convolutional layers and ReLU in attention layers [41]. Unfortunately, it is difficult to design these strategies manually, and so practitioners often use a single activation function for simplicity.

The techniques proposed in this paper may provide an avenue toward optimizing multiple activation functions in tandem. For example, the features for multiple candidate activation functions could be concatenated into a single feature vector, and this vector could be projected with UMAP to a low-dimensional space where performance prediction is more straightforward.

**Optimizing Parametric Activation Functions**    Parametric activation functions have learnable parameters that allow them to refine their shape via gradient descent. In some tasks, this extra flexibility results in better performance over fixed activation functions [5]. The techniques introduced in this paper can be readily extended to optimizing the design of parametric activation functions as well. Because the surrogate considers the state of the network and activation function at initialization, it is possible to predict the performance by treating the activation function parameters as fixed to their initial values.

However, it may be possible to extend this idea further. Because the activation function parameters are implemented as neural network weights, each parameter will have a corresponding FIM eigenvalue. These extra eigenvalues will provide the surrogate with additional information that may help predict the performance more accurately.

For simplicity, current parametric activation functions usually initialize their parameters either to be 1.0 or to approximate some existing activation function, and the initialization is usually the same throughout the network. This method is likely suboptimal; the surrogate introduced in this paper could provide a smarter approach. By adjusting the initial parameter values and observing the change in predicted performance, the surrogate can be used to find better initializations, including different ones at different layers in the network. This contribution could make parametric activation functions even more powerful.

**Optimizing Other Aspects of Neural Network Design**    By fixing the neural network architecture and varying the activation function, this paper showed that it is possible to use FIM eigenvalues to infer future performance. As the FIM is a fundamental quantity in machine learning, it may be possible to apply a similar strategy to optimize other aspects of neural network design, such as normalization layers, loss functions, or data augmentation strategies [8, 17, 18, 35]. If a meaningful distance metric between such objects can be defined, then UMAP could be used to map them to a low-dimensional space where performance prediction is much simpler.

Similarly, one could use the FIM eigenvalues to optimize alternate objectives beyond accuracy. Robustness is a particularly interesting objective, because the FIM can be used to describe a neural

network's robustness to small parameter perturbations. Other objectives, such as interpretability, fairness, or inference cost, could also be considered. For example, one could consider a multidimensional regression approach where instead of just predicting accuracy, the surrogate would predict each of these quantities separately. Such a method could present the user with a Pareto front of activation functions involving tradeoffs between these quantities.

**Reverse Engineering Activation Functions**    UMAP was used to project activation functions to a low-dimensional space, and regression algorithms to predict the performance of activation functions in this space, i.e. to serve as a fitness function for the search. However, it is possible that there is no activation function that maps to the optimum of this fitness landscape. Indeed, because such search spaces are finite, the activation functions do not completely fill them. For example, there are empty regions in Figure 4, corresponding to activation functions outside of the predefined search space.

What should be done if an empty region of the embedding space has a higher predicted fitness than any of the candidate activation functions? In the paper, these regions were simply ignored, and the activation function with the highest predicted fitness was used. However, in the future, it may be possible to create activation functions that map to these empty spaces, an in so doing improve performance. One approach could be based on inverse transforms: Given a coordinate in the low-dimensional embedding space, UMAP can apply an inverse transform and return an object that would have mapped to those coordinates. This technique was already used for visualization in Figure 3. Using this approach, UMAP could generate a hypothetical desired FIM eigenvalue distribution, or a list of activation function outputs.

There are two challenges to this approach. First, because UMAP is a dimensionality-reduction algorithm, different activation functions can map to the same location in the embedding space. Thus, the mapping from embedding space back to activation functions is not well defined. Second, even if UMAP prescribes a FIM eigenvalue distribution that is predicted to result in good performance, it may be difficult to manually design an activation function to satisfy that distribution.

However, a generated list of prescribed activation function outputs is already a good start. From this list, it is possible to construct an activation function that interpolates through these points, either in a piecewise linear fashion, with splines, or using some other standard technique. Even without the corresponding FIM eigenvalues, such an approach could potentially improve the efficiency of novel activation function discovery, and lead to better designs for activation functions in the future.

## G    Compute Infrastructure

The experiments in this paper were implemented using an AWS `g5.48xlarge` instance with eight NVIDIA A10G GPUs. The total compute cost for the search experiments in Section 6 was 14.49 GPU-hours for All-CNN-C on CIFAR-100, 21.67 GPU-hours for ResNet-56 on CIFAR-100, and 196.25 GPU-days for MobileViTv2-0.5 on ImageNet. This cost includes the time to train the eight baseline activation functions and then to evaluate 100 additional functions. The instance ran in Oregon (`us-west-2`) and was powered by renewable energy, so the experiments for this paper contributed no carbon emissions.

