# OpenReview forum: "Efficient Activation Function Optimization through Surrogate Modeling"
_NeurIPS.cc/2023/Conference — NeurIPS 2023 poster_

### Official Review · Reviewer_sps3 · 2023-07-02

**Soundness:** 4 excellent
**Presentation:** 4 excellent
**Contribution:** 3 good
**Rating:** 8
**Confidence:** 4

**Summary:**

The paper presents a new method for improving the performance of neural networks through the design of optimal activation functions. The authors created benchmark datasets by training convolutional, residual, and vision transformer architectures with systematically generated activation functions. They then developed a new surrogate-based method for optimization, which uses the spectrum of the Fisher information matrix and the activation function's output distribution to predict performance. The method was tested on CIFAR-100 and ImageNet tasks, and the results showed significant improvements in accuracy.



**Strengths:**

1. This paper introduces an innovative approach to enhancing activation functions, surpassing existing techniques in both efficiency and effectiveness.

2. The paper is exceptionally well-written, and the experiments conducted are notably thorough.

3. The benchmark datasets created by the authors provide a foundation for future research on activation function properties and their impact on performance.

**Weaknesses:**

None

**Questions:**

None

**Limitations:**

Yes

---

> ### Author Rebuttal · Authors · 2023-08-09
>
> **Response to Reviewer sps3**
>
> Thank you for the review.  Please let us know if you have any questions that we can address in the upcoming author-reviewer discussion period.

---

### Official Review · Reviewer_x7yQ · 2023-07-06

**Soundness:** 3 good
**Presentation:** 4 excellent
**Contribution:** 3 good
**Rating:** 7
**Confidence:** 4

**Summary:**

This paper introduces three benchmark datasets created by training CNN, ResNet, and ViT architectures using a set of activation functions generated from a three-node computation graph that combines unary and binary operations.

The benchmarks serve to showcase the efficacy of utilizing the 2D UMAP of the Fisher information matrix (FIM) spectrum and/or activation outputs as a cost-effective surrogate for predicting activation performance. Leveraging the 2D feature representation, an efficient activation optimization method, AQuaSurF, is developed by employing regression techniques to model activation accuracy across the 2D feature space, requiring only 100 function evaluations. The benchmark results further demonstrate the effectiveness and statistical reliability of this approach.

The proposed method is successfully applied to various vision tasks, where the discovered activation functions consistently outperform existing baseline activations. Moreover, the top activations identified through this search exhibit successful transferability to a new vision task.

**Strengths:**

The paper is well-written and easy to follow.

The approach of utilizing the UMAP embedding of the FIM spectrum with activation outputs to assess activation performance is novel and interesting.

In contrast to previous methodologies that relied mostly on evolutionary algorithms and required thousands of function evaluations, the method proposed in this work demonstrates efficiency by outperforming baselines with just 100 function evaluations.

Furthermore, the benchmark datasets introduced in this work, may potentially help accelerate research on activation optimization.

Overall, this paper offers valuable insights for assessing activation performance and also introduces a more efficient methodology for activation optimization.


**Weaknesses:**

In Section 6, the authors apply their proposed method to more challenging datasets and a larger activation search space, compared to those used to create the benchmarks. To further evaluate the effectiveness of the approach it would be beneficial to apply the method (KNR on UMAP embeddings) to vision tasks involving new network architectures as well.

While the chosen baseline activations in Table.1 already include ReLU and Swish, used in the original three architectures studied in the paper, in order to further strengthen the results it would still be advantageous, and perhaps straightforward, to extend the list of baselines at least to those used in PANGAEA, including GELU, LeakyReLU etc.

**Questions:**

1- In the first paragraph of page 3 the authors observe, based on the scatter plots in Fig 2, that "best results come from discovering functions specialized to individual tasks". However, upon comparing the upper-left and lower-right corners of the plots with the upper-right region it appears that the best functions on one task also transfer effectively to, and are potentially among the best on, the other task. Is this interpretation correct?

2- In the middle row of Fig 4. The UMAP depends only on the activations and not the model. However, there appears to be differences in the distribution of points in the 3 plots (and also compared to Fig.3). Is this because of filtering out failed activations and possible rescalings / reflections of the space? A brief comment on this would enhance clarity for readers.

3- On lines 228-229 of the manuscript "Thus, activation functions are embedded close to each other in this space if they have similar shapes, if they induce similar FIM eigenvalues, or both", considering that the metric on the union of the representations is the sum of the metrics on the individual representations, then shouldn't the activations be close to each other only if they have both similar shapes and similar FIM eigenvalues?

4- How does AQuaSurF compare with PANGAEA in terms of performance? given the partial similarity of the search spaces, is it possible to make a direct comparison between the two methods (e.g. by limiting the space to non-parametric functions)?

**Limitations:**

Limitations are partly addressed in the Future Work section in the appendix. There are no concerns regarding negative societal impact.

---

> ### Author Rebuttal · Authors · 2023-08-09
>
> **Response to Reviewer x7yQ**
>
> ---
>
> > It would be beneficial to apply the method (KNR on UMAP embeddings) to vision tasks involving new network architectures as well.
>
> This is a great idea.  We included CNN, ResNet, and ViT models in the paper to cover a wide range of possible architectures and would be happy to add additional comparisons to the revision.  We also made the AQuaSurF code publicly available and spent additional effort to write documentation and ensure that the code is easy to use, so we hope that additional comparisons in the future can be run by the community as well.
>
> ---
>
> > In order to further strengthen the results it would still be advantageous, and perhaps straightforward, to extend the list of baselines at least to those used in PANGAEA, including GELU, LeakyReLU etc.
>
> This is an excellent idea, and not only because it provides for more comparisons, but because it is not straightforward.  In order to conduct the experiment properly, we need to provide the surrogate with the performance of GELU, LeakyReLU, and any other activation functions we choose to compare against.  This extra information will naturally influence the surrogate’s predictions, and so we need to restart the search from scratch in order to make the comparison fair.  Thus, such comparisons raise an interesting question: How much does the performance of the surrogate depend on the number of initial activation functions it is given?  We are excited to run this experiment and will add it to the final revision.
>
> ---
>
> > It appears that the best functions on one task also transfer effectively to, and are potentially among the best on, the other task.
>
> Good observation.  There are indeed some activation functions that perform well across multiple architectures. However, note that the best functions are specialized to a specific task.  Note also that Figure 2 only shows the distribution of accuracies for activation functions in the benchmark datasets.  When searching in larger spaces (as was done in Section 6), we do not know what the distribution of accuracies looks like.  The most important qualitative insight from Figure 2 is that specialized activation functions do exist, and so we should exploit this fact when searching for functions in more open-ended search spaces.  We will clarify this point in the main text.
>
> ---
>
> > In the middle row of Fig 4. The UMAP depends only on the activations and not the model. However, there appears to be differences in the distribution of points in the 3 plots (and also compared to Fig.3). Is this because of filtering out failed activations and possible rescalings / reflections of the space?
>
> Yes!  This is precisely what is happening.  Thank you for reading the paper so carefully – this is an extremely subtle point.  Indeed, the plots in the middle row of Figure 4 in principle should be the same, because they do not depend on the model.  They are different because the activation functions filtered out due to invalid eigenvalues (Figure 1) are in fact different across architectures.  Furthermore, UMAP is a stochastic algorithm, so even though there is substantial overlap in the activation functions it is embedding, the final results have small variations between them.
>
> In fact, if you look closely, you can actually see the “rescalings / reflections” of the space that you hypothesized.  In the middle row, Act-Bench-CNN and Act-Bench-ResNet are nearly perfect mirror images of each other.  You can see this in the arrangement of the overall points, but also with the embedding locations of the labeled activation functions ELU, -ELU, tanh, -tanh, abs, and -abs.  The Act-Bench-ViT plot appears different and has a few small clusters of purple points in the edges of the embedding space.  These are activation functions that were not filtered out for Act-Bench-ViT but were filtered out in the other tasks.  Indeed, if you remove these points, the Act-Bench-ViT embedding space becomes almost identical to the Act-Bench-CNN one (and is a mirror image of the Act-Bench-ResNet one).
>
> We will clarify these points in the revision. Again, thank you for reading the paper so carefully.  This is an extraordinarily good insight, and we appreciate that our hard work is being given such a careful review.
>
> ---
>
> > Considering that the metric on the union of the representations is the sum of the metrics on the individual representations, then shouldn't the activations be close to each other only if they have both similar shapes and similar FIM eigenvalues?
>
> What you are describing would correspond to an intersection of the representations, but we took a union of the representations.  So, activation functions are embedded close to each other if they have similar shapes, similar FIM eigenvalues, or both.  We tried the intersection approach but found the union of the representations to be more effective.  We will clarify this point in the main text.  (See https://umap-learn.readthedocs.io/en/latest/composing_models.html for more details.)

---

> > ### Comment · Reviewer_x7yQ · 2023-08-16
> >
> > I appreciate the Authors' response and clarifications. Incorporating these insights into the paper will definitely enhance its readability.
> >
> > Given the current state of the paper, I would keep my rating of 7. However, I believe demonstrating that the proposed method, including the choice of regression algorithm and embedding dimension 1) works on a model other than those used for the benchmarks, and especially that 2) the method can discover activation functions that outperform other baseline activations, even if by adding the baseline activation to the list of initial activations, would further demonstrate the strength of the method and improve the quality of the paper.
> >
> > Regarding the generalizability concern raised by reviewer UeiT, I respectfully hold a different perspective. Activation functions are part of the network architecture which can be tailored by human experts for a particular task, just like any hyperparameter which is optimized on a validation set, and therefore this shouldn't be considered as overfitting.

---

### Official Review · Reviewer_UeiT · 2023-07-06

**Soundness:** 3 good
**Presentation:** 3 good
**Contribution:** 1 poor
**Rating:** 3
**Confidence:** 5

**Summary:**

This paper addresses the optimization of activation functions in neural networks for improved performance in machine learning tasks. The authors create benchmark datasets and propose a surrogate-based optimization method based on a characterization of the benchmark space. They apply this method to discover better activation functions in CIFAR-100 and ImageNet tasks, showcasing its practical effectiveness.

**Strengths:**

1. The authors create benchmark datasets (Act-Bench-CNN, Act-Bench-ResNet, and Act-Bench-ViT) by training various architectures with numerous activation functions.

2. The paper presents a novel surrogate-based optimization method that characterizes activation functions analytically. By utilizing the Fisher information matrix's spectrum and activation function output distribution, a low-dimensional representation is created.

3. The proposed method, AQuaSurF, efficiently discovers improved activation functions in CIFAR-100 and ImageNet tasks, surpassing previous approaches in terms of evaluation efficiency.

**Weaknesses:**

1. The motivation and definition of using "Activation Function Outputs" as feature in Section 3 is not clearly explained.

2. In Table 1, some widely used human-designed activation functions, such as ELU, ReLU, and Swish, consistently achieve top performance on various tasks with different networks. However, the top activation functions discovered by the proposed method vary across tasks and networks. This suggests a limited generalizability of the searched activation functions. In other words, when faced with a new task or utilizing a new network, the activation function needs to be searched again. Furthermore, this also implies that the searching method may overfit the specific task and network, rather than finding activation functions that are generally effective and meaningful.

3. Related to the previous point, the design of the search space appears overly complicated, which also raises concerns about overfitting. As observed, the top activation functions discovered through the search process often involve complex combinations of existing human-designed activation functions. This complexity reduces their interpretability. Human-designed activation functions, on the other hand, are typically well-reasoned and supported by theory or hypotheses, allowing them to generalize effectively across tasks and networks. However, the searched activation functions are difficult to explain in terms of why they exhibit certain characteristics, and they lack generalizability.

**Questions:**

Weaknesses 2 and 3 raise concerns regarding the significance and necessity of the proposed problem and solution. If the authors are unable to address the issues of generalizability, I would be inclined to view their "improvement" as overfitting to specific tasks and networks.

**Limitations:**

No potential negative societal impact.

---

> ### Author Rebuttal · Authors · 2023-08-09
>
> **Response to Reviewer UeiT**
>
> ---
>
> > The motivation and definition of using "Activation Function Outputs" as feature in Section 3 is not clearly explained.
>
> The intuition behind using activation function outputs as a feature is that we expect activation functions with similar shapes to have similar performance.  From one perspective, Equation 3 quantifies the difference between two activation functions’ output distributions at initialization.  But from another point of view, Equation 3 is computing the pointwise distance between two activation function shapes, giving extra weight to the middle regions near x=0 where the activation functions are more likely to be utilized.
>
> In the revision, we will explain this motivation for activation function outputs, and will clarify how Equation 3 implements this idea.  Thanks for pointing it out.
>
> ---
>
> > The top activation functions discovered by the proposed method vary across tasks and networks.
>
> General activation functions like ELU, ReLU, and Swish are useful for achieving good performance in many tasks.  However, in some tasks it is worth spending extra effort in order to achieve the absolute best performance. Customization can provide such an improvement. AQuaSurF is a way to discover customized activation functions that improve performance over the general-purpose baseline solutions in such tasks.
>
> ---
>
> > When faced with a new task or utilizing a new network, the activation function needs to be searched again.
>
> Yes, and this process allows taking advantage of customization. With previous techniques it was infeasible to perform such a new search for every task, but with AQuaSurF it is possible.  We hope that future work will build on the contributions in this paper, including the benchmark datasets and the code, and improve the efficiency even further.
>
> Note also that the best activation functions discovered often successfully transfer to new tasks and improve performance.  This is especially useful for challenging tasks such as ImageNet (Table 2).
>
> ---
>
> > This also implies that the searching method may overfit the specific task and network.
>
> Customization means finding an activation function that works as well as possible in the given context, i.e. architecture and task. The result may not work as well in another context---and that is precisely where the power of customization lies. While it is certainly possible to discover solutions that are general and apply to many contexts, they are essentially leaving money on the table. AQuaSurF provides a method for doing such customization separately for each context, thus taking advantage of any possible performance improvement.
>
> To avoid overfitting, we use standard techniques: the networks are trained on the training set, the activation functions are evaluated on a held-out validation set, and final performance is measured on the test set.
>
> ---
>
> > The top activation functions discovered through the search process often involve complex combinations of existing human-designed activation functions.
>
> This is actually an advantage of using an automated search process: It is possible to use AQuaSurF to build on any human ideas, i.e. refine and combine them, as well as augment them with entirely new designs. Such solutions can be much more complex than the original human designs; it is thus possible to discover powerful activation functions that humans are not likely to discover on their own.
>
> ---
>
> > The searched activation functions are difficult to explain in terms of why they exhibit certain characteristics.
>
> This was true of previous work like PANGAEA and Swish, but this paper actually makes key contributions in understanding what properties make an activation function effective.  The two features the surrogate model uses are informative: Activation function outputs describe how the function modifies the forward-propagated signals before training begins, and FIM eigenvalues describe the curvature of the loss surface at initialization.  The paper thus suggests that we should not limit ourselves to only using activation functions that have a simple written form---properties such as function outputs and FIM eigenvalues matter more. Based on these observations, in the future it may be possible to develop a general theory of what makes activation functions effective.
>
> To support this effort in practice, the benchmark datasets Act-Bench-CNN, Act-Bench-ResNet, and Act-Bench-ViT, as well as the AQuaSurF software, will be a powerful resource.  They already made it possible for us to identify function outputs and FIM eigenvalues as useful predictors of performance; we expect that in the future they will be useful for the community to further theoretical understanding as well as practical development of activation functions.

---

> ### Comment · Reviewer_UeiT · 2023-08-14
> **Update after rebuttal**
>
> It is unfortunate that the authors' rebuttal did not address my concerns.
>
> 1. Firstly, the author's response did not effectively address the concern regarding overfitting. If the functions found during the search on a particular task or model cannot be generalized to other tasks or models, then it constitutes a form of overfitting. This so-called "customization" lacks practical significance and does not offer new insights for academic research.
>
> 2. Secondly, taking into account the opinions of Reviewer N57C and Reviewer x7yQ, I am more inclined to agree with N57C. The improvement brought about by this costly "customization" is extremely marginal.
>
> I will keep the rating as 3. Reject.

---

> > ### Author Response · Authors · 2023-08-15
> >
> > **Response to Reviewer UeiT**
> >
> >
> > Thank you for taking the time to respond.  We strongly disagree with your assessment and have responded to each of your points below.
> >
> > First, stating that the functions “cannot be generalized to other tasks or models” is a complete misrepresentation of the paper.  Table 2 provides a direct contradiction to this statement: It shows that all nine of the activation functions discovered successfully generalized to a new task: ResNet-50 on ImageNet.
> >
> > Second, stating that customization is “overfitting” and “lacks practical significance and does not offer new insights” is patently false.  Developing custom activation functions for better performance on specific tasks is something that human researchers regularly do, and this paper provides a way to automate this design process.  Here is a concrete example: when modeling higher-order derivatives of a signal, periodic activation functions perform exceptionally well, while traditional activation functions like ReLU fail (https://arxiv.org/abs/2006.09661).  Designing an activation function with the task in mind does not constitute overfitting!  Similarly, one would not argue that CNNs have overfit to vision tasks or that RNNs have overfit to language modeling.  Rather, these are models designed to exploit task-specific structure in the data.  Our contribution is an automated method for designing activation functions that can also exploit task-specific structure to achieve better performance.
> >
> > Third, we strongly disagree that the performance improvement is “marginal.”  Our approach provided a full percentage point increase in accuracy over ReLU on four different tasks (Tables 1 and 2).  This performance improvement is on par with other work in the literature, and it is substantial given that so much effort has already gone into optimizing models for CIFAR-100 and ImageNet.
> >
> > Again, we appreciate your time in reviewing our paper, but many of the points you made contradict the facts in the paper.  Thus, we hope you that will reconsider your point of view.

---

### Official Review · Reviewer_N57C · 2023-07-11

**Soundness:** 3 good
**Presentation:** 3 good
**Contribution:** 2 fair
**Rating:** 4
**Confidence:** 3

**Summary:**

This paper introduces a set of benchmark datasets for activation function search, and an efficient search method based on the analysis of the benchmarks.


**Strengths:**

1. The proposed benchmark datasets are beneficial for further research.

2. The method that searches activation functions through the function outputs of a limited number of samples seems effective and can significantly outperform the random search baseline.


**Weaknesses:**

1. The paper is hard to follow. The main text refers to many details in Appendix, but it is still complex and hard to get to the method. I suggest the authors refine the structure and make the technical details of the proposed method more clearly.

2. Analysis is limited to show the efficiency of the method. The paper includes "Efficient" in the title, but I can only find the evaluation of efficiency in Appendix, and it should be compared with previous search methods to show how efficient it is. Besides, this method still needs to train multiple activation functions independently, which is also computationally expensive.

3. The improvements are marginal. The authors should compare their method with existing approaches in both benchmark search and new tasks search. Besides, in Table 1, comparing with the popular searched activation Swish, the improvements are marginal.


**Questions:**

None

---

> ### Author Rebuttal · Authors · 2023-08-09
>
> **Response to Reviewer N57C**
>
> ---
>
> > I suggest the authors refine the structure and make the technical details of the proposed method more clearly.
>
> Thanks for the suggestion.  Many of these details are currently in the appendix.  We will use the extra page in the camera-ready version to include more of them and to refine the structure.
>
> ---
>
> > Analysis is limited to show the efficiency of the method…it should be compared with previous search methods to show how efficient it is.
>
> To clarify, the efficiency comes from the number of evaluations needed to find a good function.  Previous approaches like PANGAEA and the algorithm that discovered Swish evaluated thousands of activation functions before discovering the best ones.  In contrast,     AQuaSurF is orders of magnitude more efficient.
>
> Figures 6 and 7 demonstrate this efficiency.  In particular, Figure 7 shows how AQuaSurF outperforms all baseline functions on ResNet-56 in just the second function evaluation.  We will revise the surrounding text to make these points clear.
>
> Because the training setups and hyperparameters are not the same, the results from previous search methods are not directly comparable.  However, a comparison can still be made by looking at the relative improvements gained by using a discovered activation function instead of ReLU (calculated as (new_acc - relu_acc) / relu_acc).  For example, with ResNet-56 on CIFAR-100 AQuaSurF results in a relative improvement of 1.65%, and in the same scenario PANGAEA gives a relative improvement of 1.64%.  Thus, the two methods discover similarly effective activation functions, but AQuaSurF does so much more efficiently: requiring 100 function evaluations instead of 1,000.
>
> ---
>
> > Besides, this method still needs to train multiple activation functions independently, which is also computationally expensive.
>
> Yes, but in some domains, it is well worth it: Spending additional compute to improve performance even a small amount may translate to significant money saved or lives improved.  The important contribution here is that while previous methods required access to distributed computing environments, AQuaSurF can be run on a single commodity cloud instance.  Previously, only for well-resourced labs were able to take advantage of activation function optimization; now it is possible for everyday practitioners, with many more applications benefiting from it.
>
> Furthermore, this paper released three activation function benchmark datasets: Act-Bench-CNN, Act-Bench-ResNet, and Act-Bench-ViT.  These resources make it possible to run search algorithms without a GPU at all.  We expect these benchmarks to be a valuable resource for the community, enabling future work to improve efficiency even further.
>
> Finally, recall that the activation functions discovered can be transferred to new tasks, even challenging ones such as ImageNet (Table 2).  Even though the best performance comes from customizing activation functions to specific tasks, these activation functions can still improve performance in other domains.
>
> ---
>
>
> > Comparing with the popular searched activation Swish, the improvements are marginal.
>
> Swish can be seen as a state-of-the-art activation function, resulting from a significant effort to optimize activation functions.  Thus, even a small improvement over Swish is significant.  In some domains like medical diagnosis or stock trading, such small improvements can make a meaningful difference.
>
> Moreover, Swish was developed in the context of tasks and architectures popular at the time, and it may not work as well in other contexts (as was already demonstrated with ResNet-v1-56 in Table 2 of the PANGAEA paper https://arxiv.org/pdf/2006.03179.pdf).  It is thus not the only activation function we will ever need; instead, it is important to be able to reliably and automatically discover better activation functions for any task and architecture that may come up in the future.  AQuaSurF’s sample efficiency will make it possible to improve performance in such new contexts as they arise.

---

> > ### Comment · Reviewer_N57C · 2023-08-20
> >
> > Thanks for the response to my proposed questions. After reading them, part of my concerns are resolved. However, I do not see more competing results or explanations during the rebuttal phase. I am worried about the quite marginal improvement against the human-designed baselines, on which I agree with Reviewer UeiT and whether it is up to the standard of NeurIPS. Besides, apple-2-apple comparison in the empirical setting is important for the NAS community, which is also a weakness in this manuscript. With this regard, I tend to keep my original rating.

---

### Author Rebuttal · Authors · 2023-08-09

**Additional Response to Reviewer x7yQ**

---

> How does AQuaSurF compare with PANGAEA in terms of performance? given the partial similarity of the search spaces, is it possible to make a direct comparison between the two methods (e.g. by limiting the space to non-parametric functions)?

In principle this experiment can be run, but there are a number of challenges, and we do not think the results would be informative enough to justify the compute cost.

PANGAEA is expensive to run.  We could limit it to the same number of function evaluations as AQuaSurF, but then it is unlikely that it would discover anything useful (Figure 4, PANGAEA paper https://arxiv.org/pdf/2006.03179.pdf).  We could also limit PANGAEA to non-parametric functions, but in this case as well it is unlikely that PANGAEA would discover good functions (Tables 3 and 4, PANGAEA paper).  As noted in the response to Reviewer N57C, AQuaSurF gave a relative improvement of 1.65% over ReLU with ResNet-56 on CIFAR-100, while PANGAEA gave a relative improvement of 1.64%.  Thus, it appears that the two algorithms are similarly capable, but AQuaSurF is orders of magnitude more efficient , achieving a comparable result in 100 function evaluations instead of 1,000.

Importantly, instead of being in competition with PANGAEA, AQuaSurF can be viewed as an enhancement of it.  AQuaSurF used a similar search space as PANGAEA because it was shown to be powerful, and the main contribution was in efficiency gains, i.e., making activation function optimization so efficient that it can be run with commodity hardware as needed for new architectures and tasks.  It is likely that the surrogate model in AQuaSurF could be synergistic with other search algorithms and search spaces, and improve their efficiency as well.

---

### Decision · Program_Chairs · 2023-09-21

**Decision:**

Accept (poster)

**Comment:**

The paper has received mixed reviews. There is a consensus among reviewers that the proposed AQuaSurf for searching activation functions from a surrogate model using limited outputs (i.e. FIM & Act) is novel. Further, the paper presents a benchmark dataset for searching activation functions in different network architectures (CNN,ResNet,ViT). As weaknesses, two reviewers (UeiT & N57C) are inclined that the paper is limited in terms of generalization and effectiveness of the search results over SOTA activations (e.g. Swish,ReLU,ELU). Authors have responded in the rebuttal and attempted to alleviate the concerns raised by the reviewers on describing the methodology and experiments. While reviewers related varying outputs from AQuaSurf to lack of generalization, the authors strongly believe this in fact is the strength of the searching method which can yield improvement over SOTA activations such as Swish. However, both reviewers still hold their original ratings. On the contrary, two other reviewers (x7yQ & sps3) strongly praise the work where the consensus is the paper introducing a novel approach for searching activation functions that require much smaller number of trials (a magnitude order faster) compared to previous methods which rely on evolutionary algorithms, requiring thousands of function evaluations to achieve optimum performance. Despite several concerns raised above, the AC finds the support from reviewers on the novelty of the paper, introducing new benchmark dataset, and effectiveness of the searching mechanisms enough to merit for publication. Further, we trust the authors will make good effort to address all concerns raised in reviews/rebuttal discussions in revised manuscript, as the final decision was significantly impacted by this agreement. The decision was discussed with and approved by the SAC.